# Cryo-plasma FIB/SEM volume imaging of biological specimens

Maud Dumoux[1]*, Thomas Glen[1], Jake LR Smith[1,2], Elaine ML Ho[3], Luis MA Perdigão[3], Avery Pennington[4], Sven Klumpe[5], Neville BY Yee[3], David Andrew Farmer[4], Pui YA Lai[4], William Bowles[1,2,4], Ron Kelley[6], Jürgen M Plitzko[5], Liang Wu[1,2], Mark Basham[4], Daniel K Clare[4], C Alistair Siebert[4], Michele C Darrow[3], James H Naismith[1,2], Michael Grange[1,2]*

[1]Structural Biology, Rosalind Franklin Institute, Didcot, United Kingdom; [2]Division of Structural Biology, Wellcome Centre for Human Genetics, University of Oxford, Oxford, United Kingdom; [3]Artificial Intelligence and Informatics, Rosalind Franklin Institute, Didcot, United Kingdom; [4]Diamond Light Source, Harwell Science & Innovation Campus, Didcot, United Kingdom; [5]Research Group Cryo-EM Technology, Max Planck Institute of Biochemistry, Martinsried, Germany; [6]Materials and Structural Analysis Division, Thermo Fisher Scientific, Eindhoven, Netherlands

*For correspondence:
maud.dumoux@rfi.ac.uk (MD);
michael.grange@rfi.ac.uk (MG)

**Abstract** Serial focussed ion beam scanning electron microscopy (FIB/SEM) enables imaging and assessment of subcellular structures on the mesoscale (10 nm to 10 µm). When applied to vitrified samples, serial FIB/SEM is also a means to target specific structures in cells and tissues while maintaining constituents' hydration shells for in situ structural biology downstream. However, the application of serial FIB/SEM imaging of non-stained cryogenic biological samples is limited due to low contrast, curtaining, and charging artefacts. We address these challenges using a cryogenic plasma FIB/SEM. We evaluated the choice of plasma ion source and imaging regimes to produce high-quality SEM images of a range of different biological samples. Using an automated workflow we produced three-dimensional volumes of bacteria, human cells, and tissue, and calculated estimates for their resolution, typically achieving 20–50 nm. Additionally, a tag-free localisation tool for regions of interest is needed to drive the application of in situ structural biology towards tissue. The combination of serial FIB/SEM with plasma-based ion sources promises a framework for targeting specific features in bulk-frozen samples (>100 µm) to produce lamellae for cryogenic electron tomography.

## Editor's evaluation

This important work is of interest to the electron microscopy and cell biology communities. The field has long searched for a suitable method to combine the pristine preservation of vitrified samples with a volumetric imaging modality that reveals subcellular architecture at sufficient contrast for ultrastructural analyses. The authors describe here the use of novel ion beams for imaging cellular samples in three dimensions, concluding that one of the four plasma sources tested produces the highest quality images, allowing them to provide several recommendations for imaging conditions along with software for improving collected images. This approach should be very useful for addressing many biological questions.

## Introduction

Volumetric imaging using a dual-beam focussed ion beam/scanning electron microscope (FIB/SEM) enables large volumes of material to be reconstructed into three-dimensional (3D) contextual maps,

for cells and tissues. However, this technique is almost exclusively applied to fixed, resin-embedded, and stained samples. There are only a few studies demonstrating application of this technique to vitrified, frozen-hydrated material (*Scher et al., 2021*; *Schertel et al., 2013*; *Zhu et al., 2021*). Vitrified (as opposed to fixed) samples are required where volumetric information is to be used for further in situ structural studies by cryogenic electron tomography (cryo-ET). Cryo-ET can determine protein structures in the context of a cell at pseudo atomic resolution but requires locating regions of interest (ROIs) within the experimental specimen (known as a lamella) sufficiently thin to be transparent to transmitted electrons (<300 nm). Lamellae are most often fabricated with cryo focussed ion beam scanning electron microscopes (cryo-FIB/SEM) (*Bäuerlein and Baumeister, 2021*). Combining ROI targeting and serial FIB/SEM analysis of biological volumes is an attractive approach to multi-scale imaging.

The process of thinning (milling) requires a focussed ion beam; the current state of the art uses Ga$^+$ ion beams. For lamella preparation, ensuring the ROI is contained within the lamella commonly employs correlative light microscopy (cryo-CLEM) relying on fluorescent markers (usually a tagged protein) prior to, and during milling (*DeRosier, 2021*; *Klein et al., 2021*; *Mahamid et al., 2015*). However, fluorescence microscopy under cryogenic conditions suffers from poor axial resolution (>300 nm), reducing its effectiveness as a targeting tool for lamella preparation. Moreover, the shuttling between machines can introduce alignment/handling errors. As structural biology moves to human tissue, developing approaches that do not depend on fluorescence to identify ROIs will be critical. An idealised approach would identify structural features in real time during milling and thus enable localisation of the ROI within 50 nm (thus ensuring the ROI is contained even in the thinnest lamella).

There are two alternatives to cryogenic fluorescence imaging: cryo-soft-X-ray tomography (*Kounatidis et al., 2020*) and serial FIB/SEM (*Schertel et al., 2013*; *Spehner et al., 2020*). Cryo-soft-X-ray tomography can image volumes up to 10 μm thick at 40 nm resolution or 1 μm thick at 25 nm and has been successfully combined with fluorescence microscopy (*Kounatidis et al., 2020*). However, soft X-ray tomography requires a synchrotron and the resulting X-ray-induced beam damage can adversely affect the sample, damaging higher resolution features and precluding cryo-ET experiments.

Serial FIB/SEM is a well-known volumetric imaging technique: a layer from the sample surface is removed by the FIB and the newly exposed surface imaged by SEM in a repeated cycle (hence serial). For this to be useful for localisation of ROI for cryo-ET, it needs to be carried out on vitrified samples. Current cryo-FIB/SEM approaches can be extended further through the introduction of an inductively coupled plasma FIB (pFIB) generated from gases as an alternative to Ga$^+$ ions. Such pFIB ion sources can be operated at higher currents than Ga$^+$ (*Burnett et al., 2016*), thus allowing sufficiently rapid ablation rates needed to remove the large volumes of material found in tissue samples (as opposed to single cells). A potential further advantage is a reduction in ion implantation into the sample which is known to occur when Ga$^+$ FIB is used with solid-state chemical specimens (*Eder et al., 2021*).

Imaging native contrast of biological material using serial FIB/SEM is difficult. The interaction with low atomic number elements results in fewer backscattered and consequently secondary electrons (*Reimer and Tollkamp, 1980*) and therefore images have inherently low sample contrast (*Reimer and Tollkamp, 1980*). Sample contrast has traditionally been improved by sample fixation and staining with heavy metal. However, this leads to ultrastructural changes and dehydration, such that constituent proteins cannot be analysed to near-atomic resolution. Therefore, increasing the contrast generated from native samples is important to extend its application to a cryo-ET workflow. For insulating materials, such as biological materials, there are two primary electron energies where the secondary electron yield is equal to the number of incident electrons – so called crossover energies. Imaging at these energies stabilises the potential of the surface and reduces charging (*Joy and Joy, 1996*; *Seiler, 1983*). Therefore, implementing acquisition schemes that utilise these crossover energies, while imaging with a short working distance electrostatic lens, may lead to greater contrast and suitability of serial FIB/SEM for targeting.

Here, we report serial cryogenic plasma FIB/SEM (cryo-serial pFIB/SEM) analysis of vitrified hydrated biological specimens. We evaluated different plasma gases for their suitability and obtained serially generated volumes with high biological contrast. We describe an automated data collection routine that maximises image contrast and feature localisation. Using this approach we imaged a range of samples from bacteria, cells, and tissue demonstrating the ability of the technique applied to

vitrified material to generate subcellular information that may be suitable for targeting. For photosynthetic bacteria, we were able to observe nascent chromatophores, in mammalian cells we quantified the extent of membrane contact sites (MCS) and mitochondrial network, in vitrified mouse brain we observed synapses and synaptic contents, while in mouse heart we were able to observe the cellular organisation of sarcomeres and other organelles within the tissue.

## Results

### Plasma FIB characterisation for FIB/SEM on plunge-frozen samples

SEM imaging is extremely sensitive to variations in surface topography, with even small changes giving rise to features parallel to the milling direction due to differential milling (known as curtains which appear as vertical lines on an SEM image). These features obscure the underlying structure and thus their presence should be minimised. We assessed the degree to which curtaining occurred on vitreous cellular samples during milling with plasma operating at different currents (*Figure 1*). We determined the curtaining propensity (expressed as % score) at three ion currents (high, medium, and low) to analyse this relationship for four different gases (argon, xenon, nitrogen, and oxygen) at two different voltages (30 and 20 kV) (*Figure 1*; *Figure 1—figure supplements 1–3*). Representative images and associated currents are shown in *Figure 1* and *Figure 1—figure supplements 2 and 3*. Variations in the curtaining showed a trend towards greater curtaining propensity at high currents vs. low currents. The notable exception to this is nitrogen at 20 kV (*Figure 1—figure supplement 2*). Nitrogen and oxygen plasmas exhibit the greatest proportion of curtaining artefacts compared to xenon and argon at high, medium, and low currents. Plasmas generated from xenon and argon produce the smoothest surfaces for bulk milling and pFIB/SEM imaging (*Figure 1*). Both argon and xenon have a curtaining propensity under 10% at low (around 0.2 nA) pFIB/SEM imaging currents while argon shows a lower propensity for curtaining at greater currents than xenon during bulk milling. During automated serial pFIB/SEM on cells and tissues we did not observe any trends in the accumulation of curtains or detectable changes in curtaining propensity over time.

Argon and xenon are monatomic gases which give rise to single plasma species ($Ar^+$ and $Xe^+$) whereas nitrogen and oxygen are diatomic gases producing multiple ion species ($O^+$, $O_2^+$ and $N^+$, $N_2^+$) which can lead to the formation of two images which partially overlap. For oxygen a double image is always formed irrespective of the imaging mode. With nitrogen, we observed the formation of double images when imaging in 'immersion' mode (*Figure 1—figure supplement 4*). This double image caused issues tracking the milled surface during serial pFIB/SEM, limiting the volume that could be obtained and leading to abortive runs. The double image can be compensated for via an extra lens in the SEM column allowing corrected FIB images to be acquired, however, we found that this compensation had to be manually varied during the experiment.

We selected argon as the optimal gas to perform serial pFIB/SEM and we performed most of our data acquisition with it. For completeness we present data using nitrogen plasma, which exhibits double imaging and has a higher propensity for curtaining.

### Plasma ion beam characterisation

The profile of the ion beam partly defines the resolution achievable during the milling step and the completeness of milling to a depth. Therefore, we characterised the beam profile of the four available gases (*Figure 1—figure supplement 5*) by measuring the full width half maximum of a 'spot burn' (see Materials and methods) at three different beam currents (high, medium, and low) at either 20 or 30 kV. As expected, the smallest probe size is considerably larger than the one reported for Ga-based source for the corresponding current (*Figure 1—figure supplement 5*; *Vitale and Sugar, 2022*). At 30 kV, we observed that probe sizes were smallest with argon, with xenon, nitrogen, and oxygen all being similar. Interestingly at 20 kV, the spot size for the different sources used are similar with argon and xenon showing an early plateau while oxygen and nitrogen increased linearly.

### Initial imaging of cells and tissue

For our first experiments we used the machine in the default configuration where the SEM image is formed at 52° relative to the FIB milling plane. We analysed human HeLa cells and intact tissue from the mouse (*Mus musculus*) brain cortex. We used a milling step of 50 nm. This step depth was chosen

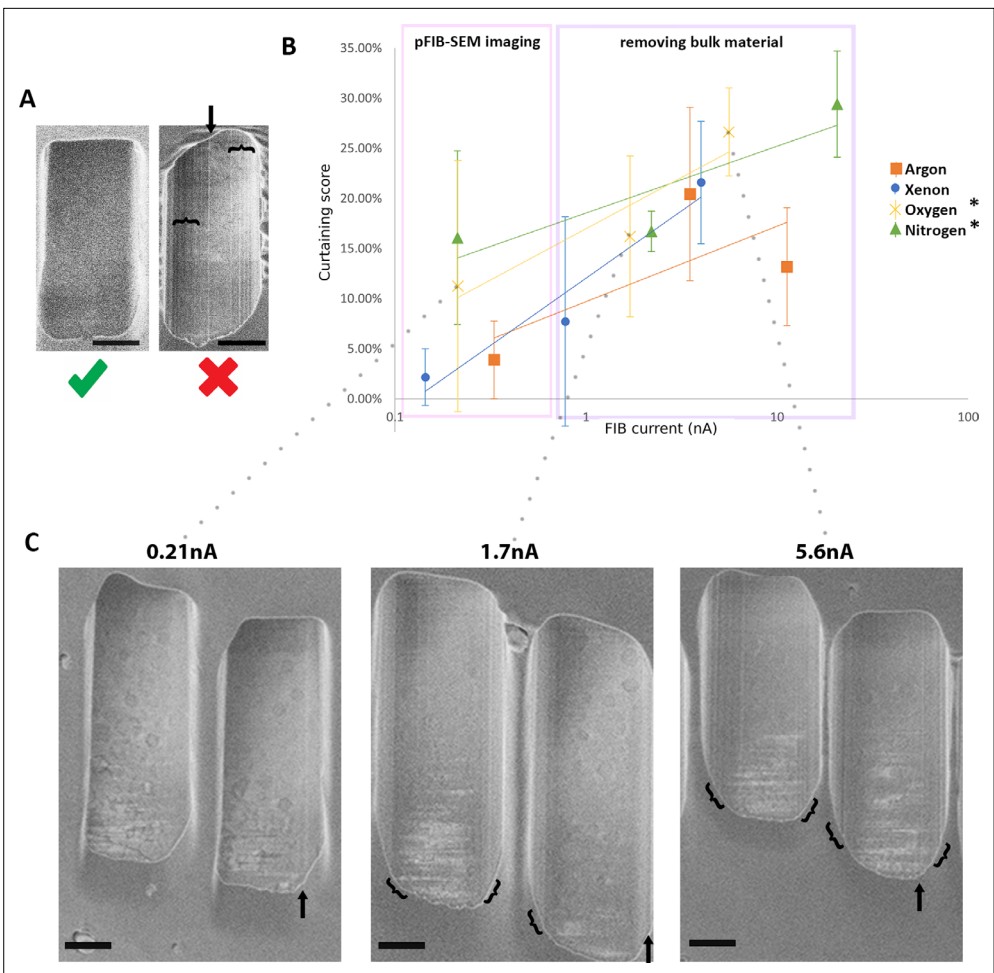

**Figure 1.** The curtaining score (higher is greater [worse] curtaining) for the different plasma sources at different currents. Plunge-frozen *Chlamydia trachomatis*-infected HeLa cells were milled for each current and each gas. 15 windows of 2 × 2.5 × 2 µm³ were milled at different measured currents at 30 kV acceleration voltage for xenon, oxygen, and nitrogen and 20 kV for argon gas (see *Figure 1—figure supplements 2 and 3*). The incidence angle of the plasma was 18°. SEM images were acquired 90° to the focussed ion beam (FIB). (**A**) Representative images from these data, showing little or no curtaining (left, oxygen, 213 pA) and extensive curtaining (right, nitrogen, 2.2 nA) seen as vertical lines. Arrow and curly brackets indicate the position of a curtain or group of curtains. Scale bar: 1 µm. (**B**) Plot of curtaining score as a function of current (see Materials and methods). Points represent the mean value associated with the standard error. The solid line represents the trend across the datapoints. n=15 per condition. (**C**) Representative images of windows generated with oxygen. Arrow and curly brackets as (A). Scale bar: 1 µm.

The online version of this article includes the following source data and figure supplement(s) for figure 1:

**Figure supplement 1.** Methods to determine the curtaining score (Materials and methods).

**Figure supplement 2.** Curtaining score for the different gases at 20 or 30 kV.

**Figure supplement 3.** Example of curtaining effects obtained at different current for (top to bottom) xenon, nitrogen, argon, and oxygen, respectively.

**Figure supplement 4.** Example of focussed ion beam (FIB) images acquired using nitrogen as ion source with (left) or without (right) double image compensation.

**Figure supplement 5.** Beam profile characterisation.

**Figure supplement 5—source data 1.** Data provided for table in *Figure 1—figure supplement 5*.

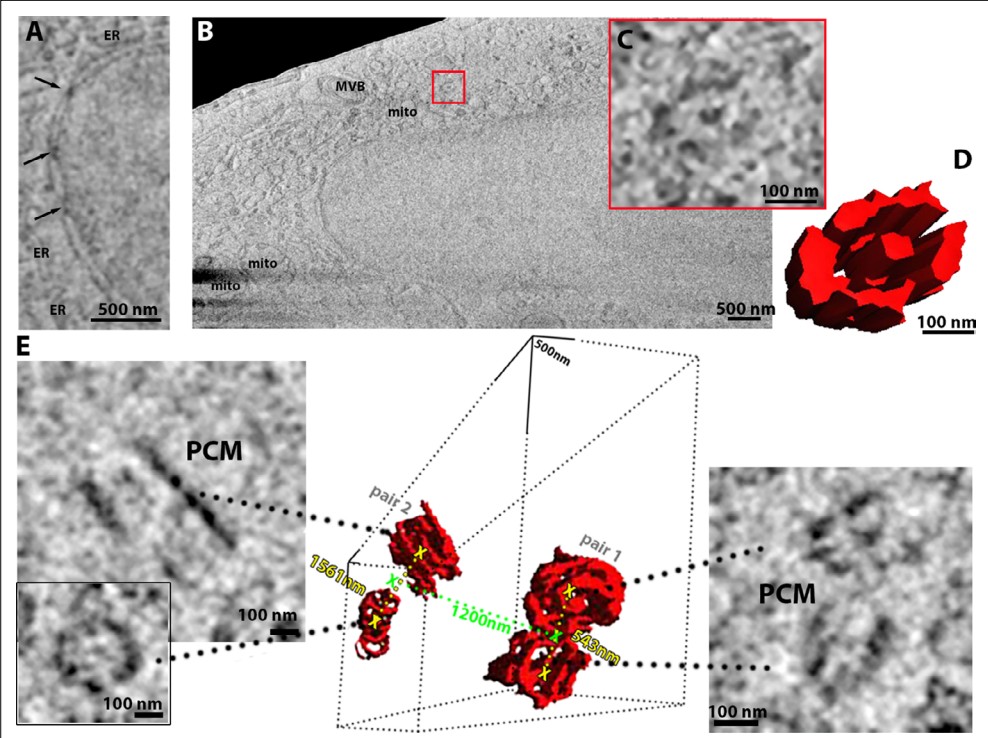

**Figure 2.** HeLa cells imaged using serial plasma focussed ion beam (pFIB)/scanning electron microscopy (SEM). (**A–E**) Serial pFIB/SEM volume acquired of a HeLa cell, using argon for milling imaged 52° to the surface by SEM. (**A**) A zoomed-in region of interest, showing nuclear pore complexes (NPCs, arrows) and endoplasmic reticulum (ER). (**B**) Overview where nucleus, mitochondria (mito), multivesicular body (MVB) and centriole (red box) are easily identifiable. (**C**) Zoom of the centriole identified in (**B**) with its three-dimensional (3D) rendering in (**D**). (**E**) This HeLa cell presented two centrosomes with two pairs of centrioles and associated pericentriolar matrix (PCM). The distance in green is between the two centrosomes respective centres and the distance in yellow depicts the distance between the centrioles in each pair. (**A–C**) Slices were filtered using a 2-pixel radius mean filter in Fiji (*Schindelin et al., 2012*). For (**E**) we used a band pass filter, also in Fiji. Full data are shown in *Figure 2—video 1*.

The online version of this article includes the following video for figure 2:

**Figure 2—video 1.** Volume from serial plasma focussed ion beam (pFIB)/scanning electron microscopy (SEM) of HeLa cells after alignment, Gaussian filtering, and cropping to the region of interest.
https://elifesciences.org/articles/83623/figures#fig2video1

based on a compromise between workflow and electron interaction depth. The interaction depth was informed by the work from Haase and colleagues (*Guehrs et al., 2017*) who estimated this to be 30 nm for entirely biological polymers and whilst imagining of silver beads inside cells reported an interaction depth of roughly 90 nm (*Seiter et al., 2014*). The focus was adjusted manually to the centre of the field of view (FOV).

The physical basis of image formation in SEM is based on the detection of secondary electrons and the contrast is formed due to the local difference of potential of different cellular constituents (*Schertel et al., 2013*). In our hands the best imaging parameters are between 1.1 and 1.2 kV (low kV), with a current between 6.3 and 13 pA (low current). We found a short dwell time (100 ns) coupled with line integration (50–100) gave optimal contrast and low charging artefact.

We recorded 500 slices on plunge-frozen HeLa cell and clearly visualised various membranous compartments, such as nucleus, endoplasmic reticulum (ER), mitochondria, and mitochondrial cristae (*Figure 2A–B*). We observed two centrosomes, recognisable by their respective pairs of centrioles characterised by their archetypal barrel/wheel shape associated to a pericentriolar matrix (*Figure 2C–E*). The distance between the centre of the two centrosomes is approximately 1.2 µm consistent with the cell in the prophase (*Figure 2E*; *Kaseda et al., 2012*). The spacing of the centrioles is 543 nm (centre to centre) in one centrosome and 1561 nm in the other, a heterogeneity not uncommon for cells in

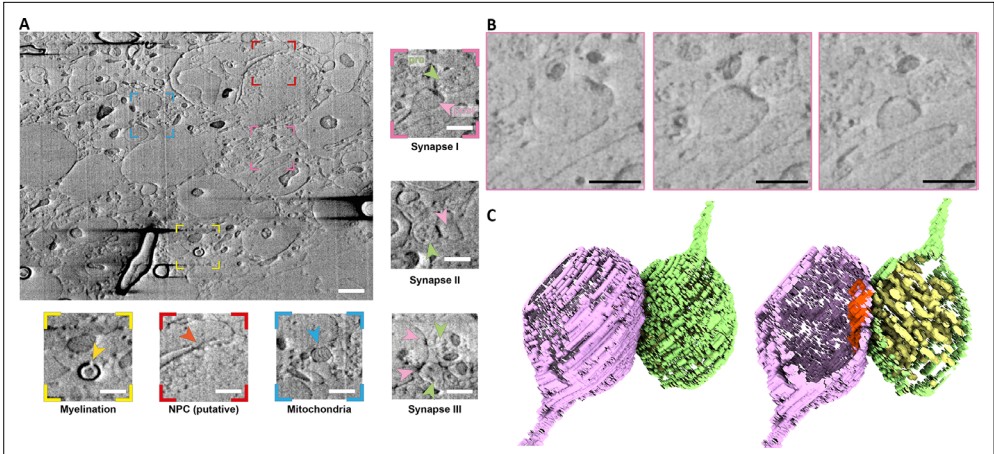

**Figure 3.** Non-fixed, high-pressure freezing (HPF) vibratome slice from a mouse brain milled using argon and scanning electron microscopy (SEM) imaging in default configuration (52° to the surface). (**A**) Representative slice of wide field of view of the serial plasma focussed ion beam (pFIB)/SEM volume. Scale bar: 1 µm. Coloured insets show regions of interest within the field of view, including myelin sheaths (yellow), putative nuclear pore complexes (red), and mitochondria (blue). Non-coloured insets show synapse morphologies from different slices. Pre- and post-synaptic cells are shown with light pink and magenta arrows, respectively. Scale bar: 500 nm. (**B**) Enlarged slices from region of (**A**) indicated in pink of a region containing a neuronal synapse with slices through Z shown at progressive positions, from left to right. Scale bar: 500 nm. (**C**) Three-dimensional (3D) volume rendering of the synapse presented in (**B**) with the pre- (green) and post- (purple) synaptic membranes labelled. The post-synaptic density (red) and pre-synaptic vesicles (yellow) are clearly identifiable. For presentation purposes the presented slices have been filtered using a 2-pixel radius mean filter. Full data are shown in *Figure 3—video 1*.

The online version of this article includes the following video for figure 3:

**Figure 3—video 1.** Volume from serial plasma focussed ion beam (pFIB)/scanning electron microscopy (SEM) of live slice of mouse brain after alignment, mean filter, and cropping to the region of interest.

https://elifesciences.org/articles/83623/figures#fig3video1

division (*Figure 2E*; *Vitiello et al., 2019*). However, the spindle microtubules were not observed between the two centrosomes.

To allow vitrification of tissue samples, high-pressure freezing (HPF) methods are used as samples are often too thick to vitrify via plunge freezing methods. HPF slices of mouse brain cortical tissue were prepared and imaged via serial pFIB/SEM (*Figure 3*). The brain tissue imaged volume was 844.4 µm³ (20.7 × 13.8 × 3 µm³) and therefore permits observation of a range of cell types including neurons and oligodendrocytes. Cellular interactions were observed, including the envelopment of axons by oligodendrocytes (visible as black contrast in images due to the high myelin [lipid] content). These features appear as elongated (oblique plane milling) or 'horseshoe' (transverse plane milling) features. The diameter of these myelinated neurons ranged from ~0.53 to ~2.8 µm (longest cross-section, mean 1.02±0.73 µm) demonstrating that these can be highly variable. Within cells we clearly observed nuclei and mitochondria (approx. 87 in the acquired volume), including the visualisation of their cristae (*Figure 3A*). Other membrane-bound compartments of varying size and diverse morphology were identified within the tissue (*Figure 3—video 1*). Synaptic junctions were visible, and we were able to discern the presence of pre-synaptic vesicles (typically 20–30 nm diameter) within the pre-synaptic terminal. Within one terminal, ~184 vesicles were identified (Synapse I, *Figure 3A–C*), whereas in a second, ~98 were identified (Synapse II, *Figure 3A*). On the post-synaptic neuron, the post-synaptic density could be visualised (*Figure 3*; *Valtschanoff and Weinberg, 2001*). Both termini connect to extruded (thinning) membranes pointing towards axons (pre) and dendrites (post). The morphologies of the synapses were very diverse, sometimes with multiple pre-synaptic terminals forming on one post-synaptic face (*Figure 3A*), highlighting the diversity prevalent in synaptic junctions in tissue. Such diversity is consistent with previous studies (*Santuy et al., 2018*).

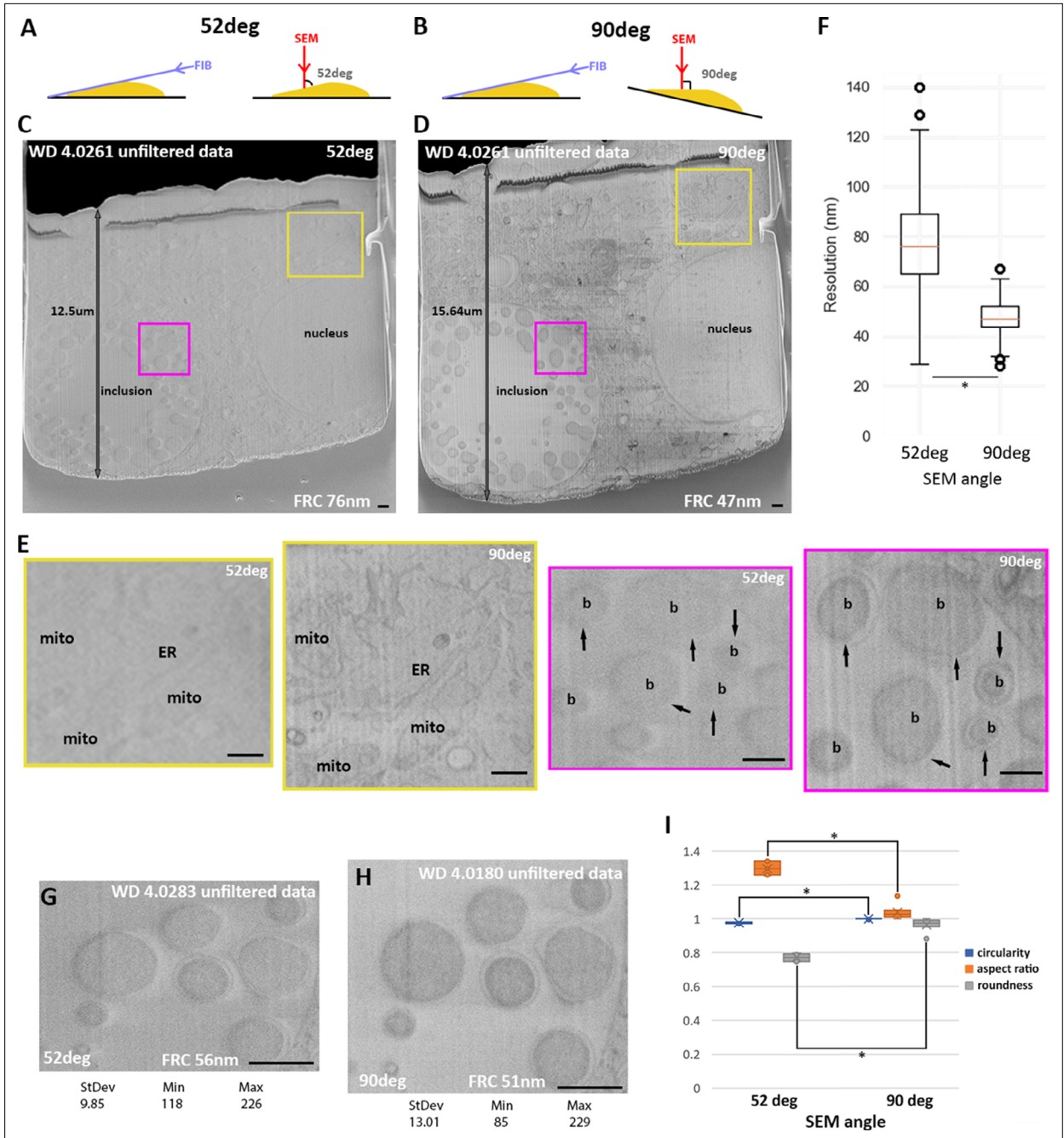

**Figure 4.** 52° vs. 90° Scanning electron microscopy (SEM) imaging angle. (**A**) and (**B**) Schematic of SEM (red arrow) and focussed ion beam (FIB) (purple arrow) angles in default and stage tilt configuration. (**A–I**) HeLa cells infected with *C. trachomatis* imaged by serial plasma focused ion beam scanning electron microscopy (pFIB)/SEM using argon to mill. (**A–E**) Images were acquired using the same parameters except for the tilt angle. No histogram modifications have been made and no filters have been applied. (**C**) and (**D**) are the full field of view and (**E**) zoomed-in panel (colour boxes). Arrows show the outer membrane of bacteria. (**F**) presents the resolution obtained for the different patches on images acquired at 52° (n=251 patches) or 90°(n=303 patches) to the surface using FRC to measure. (**G–H**) In addition to the angle, the working distance (WD) was modified while other imaging parameters kept identical. (**I**) Bacteria (n=8) were segmented for circularity, aspect ratio, and roundness to be calculated (*Rueden et al., 2017*) and presented in this graph as a function of the imaging angle with the standard deviation. All values were normalised to the case of a perfect circle having a value of 1.0. *Indicates a significant difference (0.1%). WD: working distance, FRC: Fourier ring correlation, mito: mitochondria, ER: endoplasmic reticulum, b: bacteria. Scale bars: 500 nm (**C, D and E**) and 1 μm (**G and H**).

The online version of this article includes the following figure supplement(s) for figure 4:

**Figure supplement 1.** Fourier ring correlation (FRC) evaluation method (Materials and methods).

*Figure 4 continued on next page*

*Figure 4 continued*

**Figure supplement 2.** Relationship between resolution and working distance.

**Figure supplement 3.** Measurement of *XY* (**A and B**) and *Z* drift (**C and D**) during data acquisition with no stage movement (NSM) imaged 52° to the scanning electron microscopy (SEM) or with stage movement (SM), imaged 90° to the SEM.

**Figure supplement 4.** Measurement of depth of field.

## Optimising imaging quality

We explored whether we could further improve image quality by carrying out the SEM imaging 90° to the milled sample surface. To achieve this, a compensatory stage tilt between milling and imaging steps is required (*Figure 4A and B*). To evaluate the gain in contrast introduced by this tilt, we imaged HeLa cells infected with *C. trachomatis*. These cells can be readily plunge-frozen and exhibit a wide range of subcellular features (bacteria, vacuoles, Golgi apparatus, nuclei, large protein assemblies, e.g., NPC) enabling visual analysis of contrast and information content. The samples are also large enough to enable a wide FOV. The cells were milled with argon and subsequently imaged at the two different angles. Other parameters, such as stage positions, working distance, and imaging parameters (voltage, current, line integration, exposure, detectors voltage offset, and gain) were kept constant. The contrast improvement when SEM images are taken at 90° (*Figure 4C–E*) is immediately visible, features within images acquired at 52° are less pronounced compared to imaging at 90° to the surface, with features such as the nucleus, cytoplasmic vesicles, and bacterial inclusions, while discernible in both angles, are much more prominent in images acquired at 90°. The outer membrane of the bacteria exemplifies a case where features are low contrast when imaged at 52°, but which at 90° can be prominently seen in the images (*Figure 4E* arrows). The improvement in quality would be expected to enhance image analysis and segmentation.

Imaging a tilted surface causes a shortening of features along the *y*-axis in the FOV (e.g. by a factor of $\sin(\alpha)$, assuming $\alpha$ is either 90° or 52°) (*Figure 4*). We segmented the *C. trachomatis* cell envelope to quantify the effect of 52° tilted data on the sphericity. At 52° this elongation results in cells appearing as ellipsoids (*Figure 4I*). At 90° these features are circular.

Although obvious visually, we sought to determine a more quantitative assessment of the improvement in image quality. We therefore implemented a Fourier ring correlation (FRC) approach based on the one-image FRC used in fluorescence microscopy (*Koho et al., 2019*; *Figure 4—figure supplement 1*). In one-image FRC, a single image is split into two by taking even and odd pixels in a checkerboard pattern and comparing the two. Patches of 256×256 pixels on the image were generated for assessment, producing maps of local resolution across the whole FOV (*Figure 4—figure supplement 2*).

When estimating the resolution improvement due to the difference in the angle, other imaging parameters were kept constant (*Figure 4C and D*). SEM imaging at 52° produced a global image resolution on average over a whole FOV of 76±19 nm, while using the same assessment with SEM imaging at 90° gave a resolution of 47±7 nm over the whole FOV (*Figure 4J*). Manual imaging at 52° with optimisation of focus improved the reported resolution (*Figure 4G and H* and *Figure 4—figure supplement 2*). Indeed, the bacterial outer membrane is now more visible at 52° than before. However, visual analysis of features including the outer membrane show that even with manual control, 90° imaging yields more detail and higher contrast due to differences in electron collection efficiency which is maximal when normal to the SEM (*Wells, 1975*). This could be measured observing the minimum and maximum values as well as the standard deviation of the pixel intensity values (*Figure 4G and H*).

Stage movement can introduce image shifts which in turn can introduce complexity and errors in downstream analysis. We measured the *XY* and *Z* shift using 1 μm beads that were imaged at 52° (no movement) or at 90° following stage movement. The *XY* shift was calculated as the shift between features and measured to be 52±38 nm without stage movement, and 96±71 nm with stage movement. Thus, stage movement results in twofold increase in drift, however the absolute magnitude of the drift is small, representing under 1% of the FOV (*Figure 4—figure supplement 3A, B*). This small increase in drift will have little (if any) effect on the alignment. For potential *Z* shift, we used the sphericity (and thus change in diameter with depth) of the beads to determine discrepancies between the programmed milling step size and the actual step size (derived by measuring diameters). For a targeted milling step of 50 nm, the observed steps were 48.1±1.4 nm without stage movement

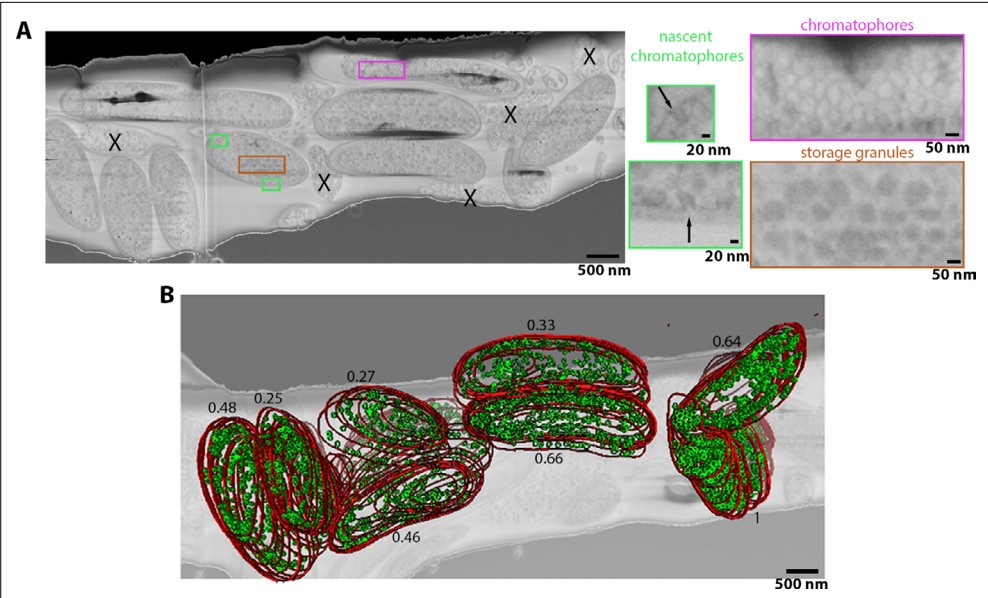

**Figure 5.** *R. rubrum* image stacks acquired using argon serial plasma focussed ion beam (pFIB)/scanning electron microscopy (SEM) normal to the SEM column. (**A**) Full field of view of slice of serial pFIB/SEM volume, showing vitrified *R. rubrum*. Features within the bacteria are shown enlarged (right), with highlighted areas showing nascent chromatophores (green), mature chromatophores (pink), and storage granules (brown). X indicates dead/dying bacteria or debris. (**B**) Slice of the volume superimposed with the volume rendering after segmentation. In red is the outer membrane and green the mature chromatophores. The number associated with each segmented bacteria is the ratio of the number of chromatophores per sum of the surfaces occupied by the bacteria at each slice. The slice was filtered using a 2-pixel radius Gaussian filter. Full data are shown in *Figure 5—video 1*.

The online version of this article includes the following video for figure 5:

**Figure 5—video 1.** Volume from serial plasma focussed ion beam (pFIB)/scanning electron microscopy (SEM) of *R. rubrum* after alignment, mean filter, and cropping to the region of interest.

https://elifesciences.org/articles/83623/figures#fig5video1

and 48.9±4 nm with stage movement. *Z*-drift is therefore minimal and not increased by the tilting regime (*Figure 4—figure supplement 3C, D*). Moreover, this experiment confirms the targeted *Z* step (50 nm) is achievable under cryogenic conditions (*Chan and Chason, 2007*).

Finalyy, the depth of field is of interest as it informs how much volume can be milled without the need for autofocus correction methods whilst retaining image quality. At both 1 and 2kV imaging energy, this was found to be ~20 μm (*Figure 4—figure supplement 4*). Using a slice thickness of 50 nm, a focus correction will be needed after around 400 slices. Fully automating focus corrections would be desirable for thicker samples. Alternatively, a focus step change based on the milling steps could be implemented as a script within the Auto Slice and View (ASV) software from Thermo Fisher.

## The cryo-serial pFIB/SEM method

We applied the 90° imaging approach to a range of samples to evaluate its potential to generate new biological insights.

### Rhodospirillum rubrum

*R. rubrum* are purple bacteria with a complex cytoplasmic organisation due to theie photosynthetic ability. Plunge-frozen *R. rubrum* were imaged in a frozen-hydrated state, absent of stain or fixative (*Figure 5*). Minimal subcellular artefacts were observed, and biological features including lipid drop-lets (LDs) and inner and outer membranes, thought to be 30–80 nm apart (*Tucker et al., 2010*), were clearly distinguished within the pFIB/SEM image stack (*Figure 5A–B*). Collection of the dataset was accomplished in 2.5 hr in an automated manner. We observed both developing and mature chromato-phores; the roughly spherical vesicles which contain the photosynthetic apparatus and are ~30–60 nm

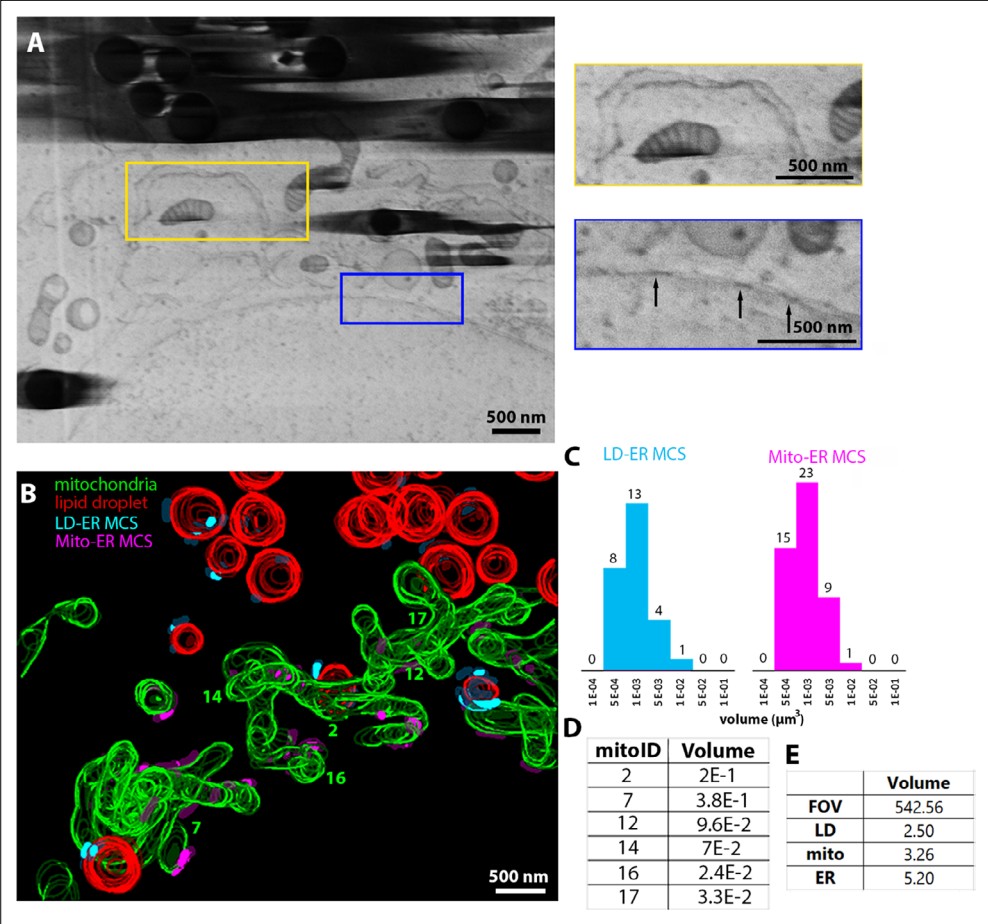

**Figure 6.** Vero cells imaged using serial plasma focussed ion beam (pFIB)/scanning electron microscopy (SEM) normal to surface. (**A–E**) Serial pFIB/SEM volume acquired of a Vero cell, using nitrogen plasma for milling. (**A**) Field of view (FOV) with insets show enlarged regions of endoplasmic reticulum (ER) and mitochondria with visible cristae (yellow), and enlarged region populated with nuclear pore complexes (blue) (arrows). The vertical curtains on the left of the yellow box in the large FOV image (left) arise from contamination at the surface of the cell. (**B**) Three-dimensional (3D) segmentation of subcellular features within the volume in (**A**). Red: lipid droplets (LDs), green: mitochondria, cyan: LD-to-ER membrane contact site (MCS) (max. 25 nm space between the membranes), purple: mitochondria to ER MCS. (**C**) Number of different ER MCS in contact with organelles vs. the contact volume ($\mu m^3$), for LDs (blue) and mitochondria (pink). (**D**) Volume ($\mu m^3$) of complete mitochondria within the volume of the Vero cell are shown. (**E**) Volume ($\mu m^3$) of the FOV and different organelles. Full data are shown in *Figure 6—video 1* and full segmentation including ER are available in *Figure 6—video 2*. Source data are provided for tables included in this figure.

The online version of this article includes the following video and source data for figure 6:

**Source data 1.** Data provided for table in *Figure 6*.

**Figure 6—video 1.** Volume from serial plasma focussed ion beam (pFIB)/scanning electron microscopy (SEM) of Vero cells after alignment, mean filter, and cropping to the region of interest.
https://elifesciences.org/articles/83623/figures#fig6video1

**Figure 6—video 2.** Segmentation (Amira) from the volume from serial plasma focussed ion beam (pFIB)/scanning electron microscopy (SEM) of Vero cells after alignment, mean filter, and cropping to the region of interest.
https://elifesciences.org/articles/83623/figures#fig6video2

in diameter when mature (*Tucker et al., 2010*). The formation of these chromatophores had previously not been visualised on this scale. Nascent chromatophores budding from the inner membrane were unambiguously discerned (*Figure 5A*). In the eight cells which we segmented, 2772 mature and 84 nascent chromatophores were observed. The segmented data comprises ~33% of the total number of cells, hence an estimated ~8000 chromatophores would be available for analysis in the full volume.

We noted that the nascent chromatophores were unevenly distributed across the population of cells; in one of the segmented cell, ~12% of the total chromatophores were nascent whereas the majority of cells imaged had very few or none. The identification of this atypical and rare cell was possible as we imaged 12 bacteria volumetrically at mesoscale resolution. The ability of serial pFIB/SEM ability to image large volumes quickly allows rare events such as chromatophore budding to be observed.

## Vero cells

Plunge-frozen vitrified Vero cells were milled using nitrogen. Nitrogen plasma was chosen to investigate whether the worst performing gas (in terms of curtaining) was still able to provide useful insights. Imaging with nitrogen severely limited the software's ability to track the milling area, this is reflected in the number of $Z$ slices acquired when compared to HeLa cells (46 slices vs. 500 for HeLa). This volume was also acquired after seven previous attempts to automate serial pFIB/SEM acquisition. However, we were able to clearly see subcellular structure. In addition to membranous organelles, we identified large protein complexes within the cellular volumes. Nuclear pore complexes (NPCs) were observed (*Figure 6A*) within cells permitting us to visualise their organisation within the cell, with 138 NPCs being visible within the data. The mitochondrial network (*Shen et al., 2022*) was also characterised (*Figure 6A, B and D*) with isolated mitochondria (mitochondria 16 and 17), small mitochondrial networks (12 and 14), and large networks (2 and 7) identifiable. We calculated the volume taken up by mitochondria, ER and LDs within the imaged volume (*Figure 6D and E*), and quantified the number and size of contacts of the ER with LDs or with mitochondria (*Figure 6A, B and E*). In Vero cells, we identified 26 LD-to-ER contacts and 48 mitochondria-to-ER MCS (*Giacomello and Pellegrini, 2016*; *Jacquemyn et al., 2017*). MCS vary in length (membrane to membrane contact) and distance between the two organelles. In our analysis, we kept a constant distance between organelles (25 nm which is the maximum separation between membranes). Therefore, the analysis is independent of the intermembrane variation and includes all MCS. Both LD-ER and mitochondria-ER MCS have a similar volume ($1-5 \times 10^{-3}$ µm³) (*Figure 6A*) and Gaussian distribution. Consequently, ER contact sites measures vary depending on how long and how spread those contacts are. Interestingly, the distribution is the same, suggesting a limit in the extent of which the ER can track and contact an organelle.

## High-pressure frozen heart tissue

We further demonstrated the use of normal (90°) imaging applied to fixed (4% paraformaldehyde) heart tissue, which was sectioned and subsequently vitrified in a similar manner as described for brain tissue. Fixation was used to preserve tissue integrity in the time between sectioning where the tissue may have degraded. Cryo-serial pFIB/SEM was then subsequently performed without further treatment. Sarcomere-containing regions (myofibrils) were visualised sandwiched between columns of mitochondria, running along and at oblique angles with respect to the milling direction (*Figure 7A*). The Z-disc, I-band, and A-bands of the sarcomeres were visible, along with a putative M-line (*Figure 7A and B*). The average sarcomere length in these samples was 1.68 µm (SD = 0.03, *N*=34). As cardiomyocytes are syncytial, they have multiple nuclei. One nucleus was seen with associated NPCs and putative hetero/euchromatin regions. Blood vessels, in proximity to the cell, were identified by the characteristic layer of endothelial cells and red blood cells (bi-lobal shape) (*Figure 7B*). In the heart volume examined (2698.4 µm³), 997 mitochondria were visualised. This gives a relative concentration of mitochondria of 0.36 mitochondria/µm³ for heart tissue (c.f. 0.10 for mouse brain), which agrees with previous observations of mitochondrial content in mouse heart tissues (*Else and Hulbert, 1985*).

## Resolution estimation of the datasets

The FRC approach was used to assess the resolution of our datasets. Interestingly, the brain section and HeLa cells, which were acquired using the same software, angle, and pixel size, present different resolutions reflecting the importance of other parameters such as focussing and astigmatism, as well as sample to sample variability. *R. rubrum* and Vero cells were also imaged using similar pixel size, identical software, and imaging angle, but present a difference in resolution (twofold in favour of the Vero cell). The FRC measurement of images at the beginning, middle, and end of each of the biological sample datasets allowed assessment as to how resolution may vary over a given serial pFIB/SEM acquisition series (*Figure 8* and *Figure 8—figure supplement 1*). We determined that in each acquisition series there was no evidence for resolution degradation during data collection,

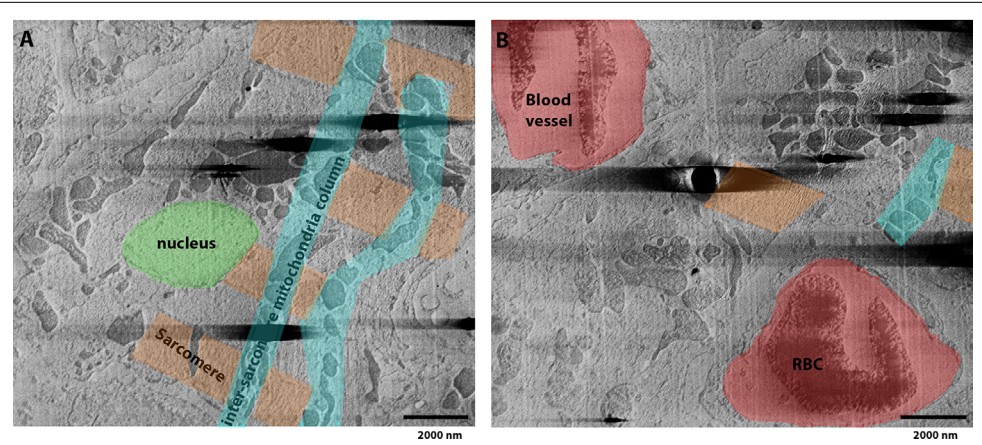

**Figure 7.** Fixed, high-pressure frozen (HPF) vibratome slice from a mouse heart image milled using argon and imaged normal to the surface. (**A**) Slice from serial plasma focussed ion beam (pFIB)/scanning electron microscope (SEM) volume showing subcellular structures consistent with those recognised from a cardiac myofibril (*Begay et al., 2018*), with a nucleus (green), sarcomeric elements (including Z-disc, I-band, and A-band) (orange), and mitochondria organised between axially organised sarcomeres in columns (cyan). (**B**) Slice of the tomogram preceding the image shown in (**A**), where amongst the cardiac tissue, blood vessels and red blood cells (RBC) can be identified (red). *Figure 7—video 1* presents the whole data.

The online version of this article includes the following video for figure 7:

**Figure 7—video 1.** Volume from serial plasma focussed ion beam (pFIB)/scanning electron microscopy (SEM) of fixed slice of mouse heart after alignment, mean filter, and cropping to the region of interest.
https://elifesciences.org/articles/83623/figures#fig7video1

even in the case of longer data acquisition runs (HeLa cells – 500 slices). In this case the software used was the ASV software (Thermo Fisher), which integrated an estimation-based focus change for each step.

## Combining pFIB/SEM with cryo-CLEM and cryo-ET

An advantage of the cryo-serial pFIB/SEM approach is the absence of chemical fixation and staining, meaning that the specimen is in principle amenable for targeted lamellae preparation for downstream cryo-ET. To validate this, we generated lamellae of *R. rubrum* and human retinal pigment epithelial cells (RPE-1) and imaged them using SEM during fabrication and transmission electron microscopy (TEM) after fabrication. The final milling steps for lamella preparation and subsequent cryo-ET analysis generally employed lower currents than those used for cryo-serial pFIB/SEM imaging (60 pA [*Berger et al., 2023*] vs. 200 pA). Two-hundred pA was initially used for serial pFIB/SEM, followed by final step of milling at 60 pA (*Figure 9—figure supplement 1* and *Figure 9—video 1*). The SEM and TEM images of the same lamella demonstrated that it was possible to correlate features, such as subcellular structures. Where a short milling step performed after SEM imaging there will be physical differences between the lamella imaged by SEM and that which was imaged by TEM for example, the size of the LD or compartments (*Figure 9—figure supplement 1*). Even without a final milling step, differences were observed (*Figure 9*; *Figure 9—figure supplement 2*). These differences arise from the distinctive physics of the techniques; SEM images regions with approximately 25 nm depth at the surface, whereas the entire thickness of the lamella (up to 300 nm) is imaged by TEM. Thus, when imaging a small organism such as *R. rubrum*, the SEM images contain only part of a bacterium whereas in TEM essentially the complete cell is visualised (*Figure 9—figure supplement 2* and *Figure 9—video 2*). Nonetheless it was straightforward to correlate the photosynthetic vesicles between the modalities. We were also able to correlate fluorescence from chlorophyll (recorded with the integral fluorescence microscope) with both pFIB/SEM and TEM images (*Figure 9* and *Figure 9—video 3*). We conclude that serial pFIB/SEM is suitable for correlative workflows at cryogenic temperature on near-native state samples.

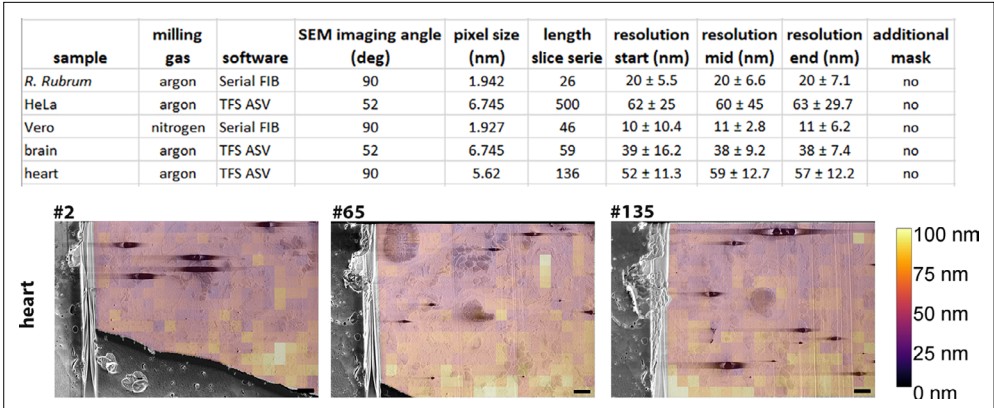

| sample | milling gas | software | SEM imaging angle (deg) | pixel size (nm) | length slice serie | resolution start (nm) | resolution mid (nm) | resolution end (nm) | additional mask |
|---|---|---|---|---|---|---|---|---|---|
| *R. Rubrum* | argon | Serial FIB | 90 | 1.942 | 26 | 20 ± 5.5 | 20 ± 6.6 | 20 ± 7.1 | no |
| HeLa | argon | TFS ASV | 52 | 6.745 | 500 | 62 ± 25 | 60 ± 45 | 63 ± 29.7 | no |
| Vero | nitrogen | Serial FIB | 90 | 1.927 | 46 | 10 ± 10.4 | 11 ± 2.8 | 11 ± 6.2 | no |
| brain | argon | TFS ASV | 52 | 6.745 | 59 | 39 ± 16.2 | 38 ± 9.2 | 38 ± 7.4 | no |
| heart | argon | TFS ASV | 90 | 5.62 | 136 | 52 ± 11.3 | 59 ± 12.7 | 57 ± 12.2 | no |

**Figure 8.** Fourier ring correlation (FRC) resolution measurements for biological samples. (**A**) Table presents the acquisition parameters and the associated calculated resolution for the dataset presented in *Figures 3–6*. The length of the series represents the number of slices. All datasets were acquired using a 50 nm step. FRC resolution measurements were determined using image slices taken from serial plasma focussed ion beam (pFIB)/scanning electron microscope (SEM) volumes - three slices were used (start, mid, end) from each dataset. In all cases, the organoplatinum layer was masked out of these measurements. Software used are either SerialFIB (*Klumpe et al., 2021*) or Thermo Fisher Scientific Auto Slice and View (TFS ASV). For *R. Rubrum* the number of patches is n=383, for HeLa n=95, for Vero n = 251, for the brain n = 95 and for the heart n = 301, 383 and 385 for the start, middle and end respectively. (**B**) Example of the FRC analysis from the heart dataset. Slice numbers 2, 65, and 135 are the one used for start, mid, and end. Scale bars: 2 µm. *Figure 8—figure supplement 1*: FRC overlays for HeLa, brain, and *R. rubrum*. Source data are provided for tables included in this figure.

The online version of this article includes the following source data and figure supplement(s) for figure 8:

**Source data 1.** Data provided for table in *Figure 8*.

**Figure supplement 1.** Analysis of the resolution from different biological data acquired at different points during data acquisition (start, middle, end).

## Mitigating charging artefacts

Charging occurs when the energetic electron beam interacts with highly insulating substances in the sample. We typically found this effect in regions associated with a greater lipid content (lipidic membranes, lipid vesicles, myelin) (*Figures 2–7*). The presence of charging manifests as intense dark spots with asymmetric streaks along the scan direction.

We developed an approach that enabled computational removal of charging artefacts in images. For example, *Saccharomyces cerevisiae* data imaged at 90° to the surface contain extensive charging of LDs. Such an approach to mitigate charging would not only characterise the effect of charging on image content but may also facilitate segmentation and image analysis. The algorithm required segmentation of the charged centres but not the asymmetric streaking using a pre-trained U-Net neural network (*Ronneberger et al., 2015*) with manual clean-up (*Figure 10A*). Comparing the auto-mated segmentation of charge centres with the automated with manual clean-up yielded a 0.93 Dice coefficient score (*Dice, 1945*; *Sørensen, 1948*). This high Dice coefficient score indicates that little manual clean-up was required and full automation of this step could be possible in future.

After charge centres had been segmented, a localised, row-by-row filter was applied which gath-ered the data near the annotated regions and compares it with the average of preceding rows. A smoothly varying continuous function was fitted and subtracted from the data line (see Materials and methods). As a result, biological features adjacent to the charging artefact were partially restored (*Figure 10B and C*). However, for regions immediately adjacent to and the charged region itself, there were no information to recover. After charge removal the overall image resolution remains unchanged.

To quantify the effects of the charge mitigation, averaged line profiles were used in both the hori-zontal and vertical directions, centred on each charging centre before and after charge mitigation (*Figure 10E and F*). For this analysis, we used isolated complete charging centres. A line profile repre-senting 9× the diameter of the charging centre was used to fully capture all effects due to charging (*Figure 10G*). Prior to charge mitigation, the average contrast of the biological material adjacent to charging centres is much diminished and the dip in the line profiles due to this is variable and broad

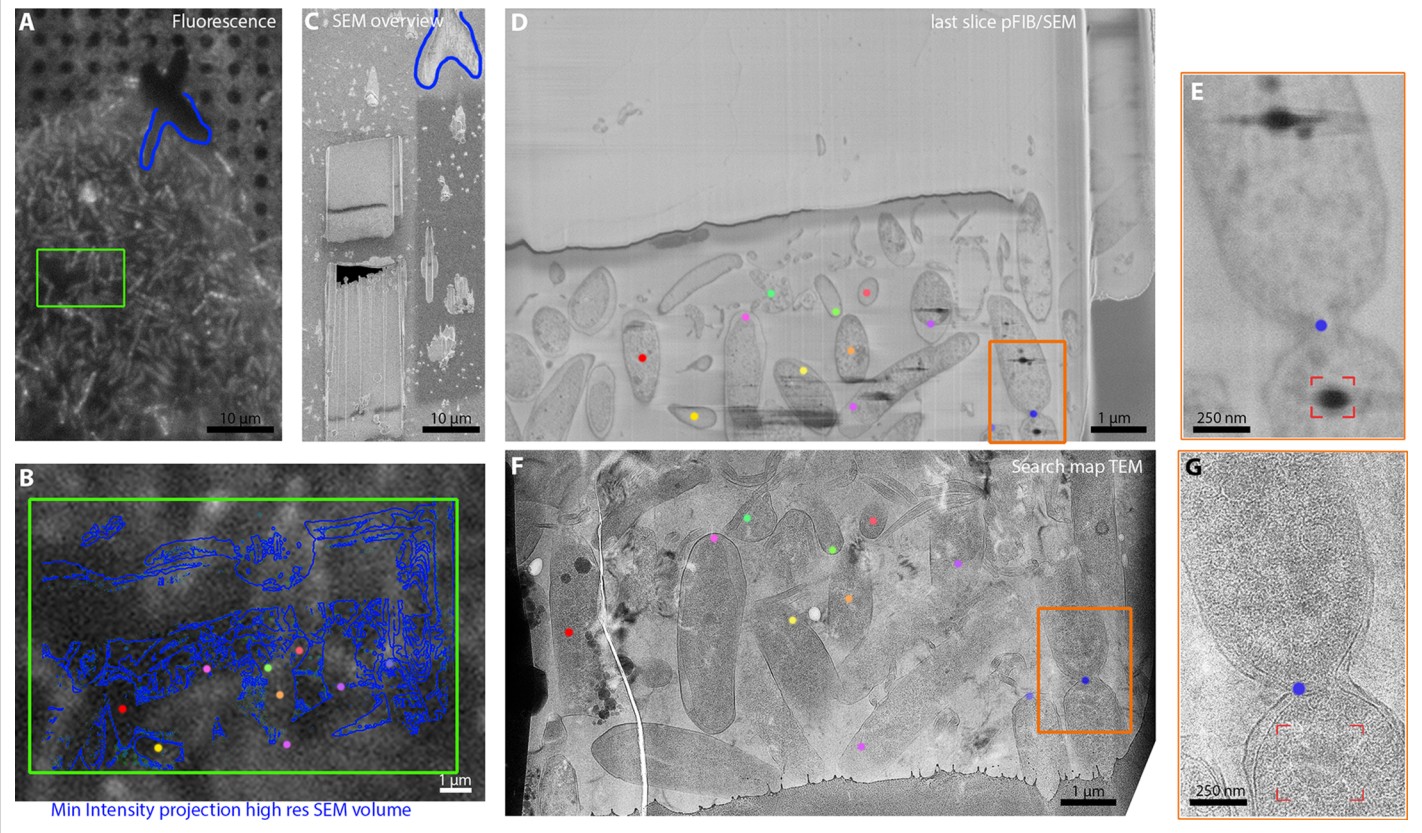

**Figure 9.** Correlation of fluorescence, serial plasma focussed ion beam (pFIB)/scanning electron microscopy (SEM), and transmitted electron microscopy (TEM) at cryogenic temperatures. (**A and B**) *R. rubrum* were imaged using fluorescence microscopy within the dual-beam microscope chamber before the deposition of the protective platinum layers. Serial pFIB/SEM was then performed (**D and E** and *Figure 9—video 1*) and then TEM images of the lamella acquired subsequently (**F and G**). The coloured dots presented in panels B and D–G are common features between the different imaging modalities. The green rectangles in A and B represent the regions of interest (ROI) where serial pFIB/SEM was acquired. The orange rectangle in D and E highlights the septum on both SEM and TEM while the red corner the presence of lipid droplets in E and G. The blue outline present on (**A–C**) indicates the outline used to align the fluorescence and SEM images. The images were not filtered.

The online version of this article includes the following video and figure supplement(s) for figure 9:

**Figure supplement 1.** Correlation between plasma focussed ion beam (pFIB)/scanning electron microscopy (SEM) and transmitted electron microscopy (TEM).

**Figure supplement 2.** Correlation between serial plasma focussed ion beam (pFIB)/scanning electron microscopy (SEM) and cryo-electron tomography.

**Figure 9—video 1.** Volume from serial plasma focussed ion beam (pFIB)/scanning electron microscopy (SEM) of fixed slice of of *R. rubrum* after alignment and cropping to the region of interest.

https://elifesciences.org/articles/83623/figures#fig9video1

**Figure 9—video 2.** Volume from serial plasma focussed ion beam (pFIB)/scanning electron microscopy (SEM) of fixed slice of of retinal pigment epithelial-1 (RPE-1) cells after alignment and cropping to the region of interest.

https://elifesciences.org/articles/83623/figures#fig9video2

**Figure 9—video 3.** Volume from serial plasma focussed ion beam (pFIB)/scanning electron microscopy (SEM) of fixed slice of of *R. rubrum* after alignment and cropping to the region of interest.

https://elifesciences.org/articles/83623/figures#fig9video3

dependent on the size and severity of the artefact. This contrasts with the vertical line profiles where a clear, sharp dip due to the LD is consistently seen with grey values indicative of biological features on either side. Upon mitigation of the charging artefacts, a similar profile can be recovered, though with some asymmetry to the grey values still present on either side of the charge centre.

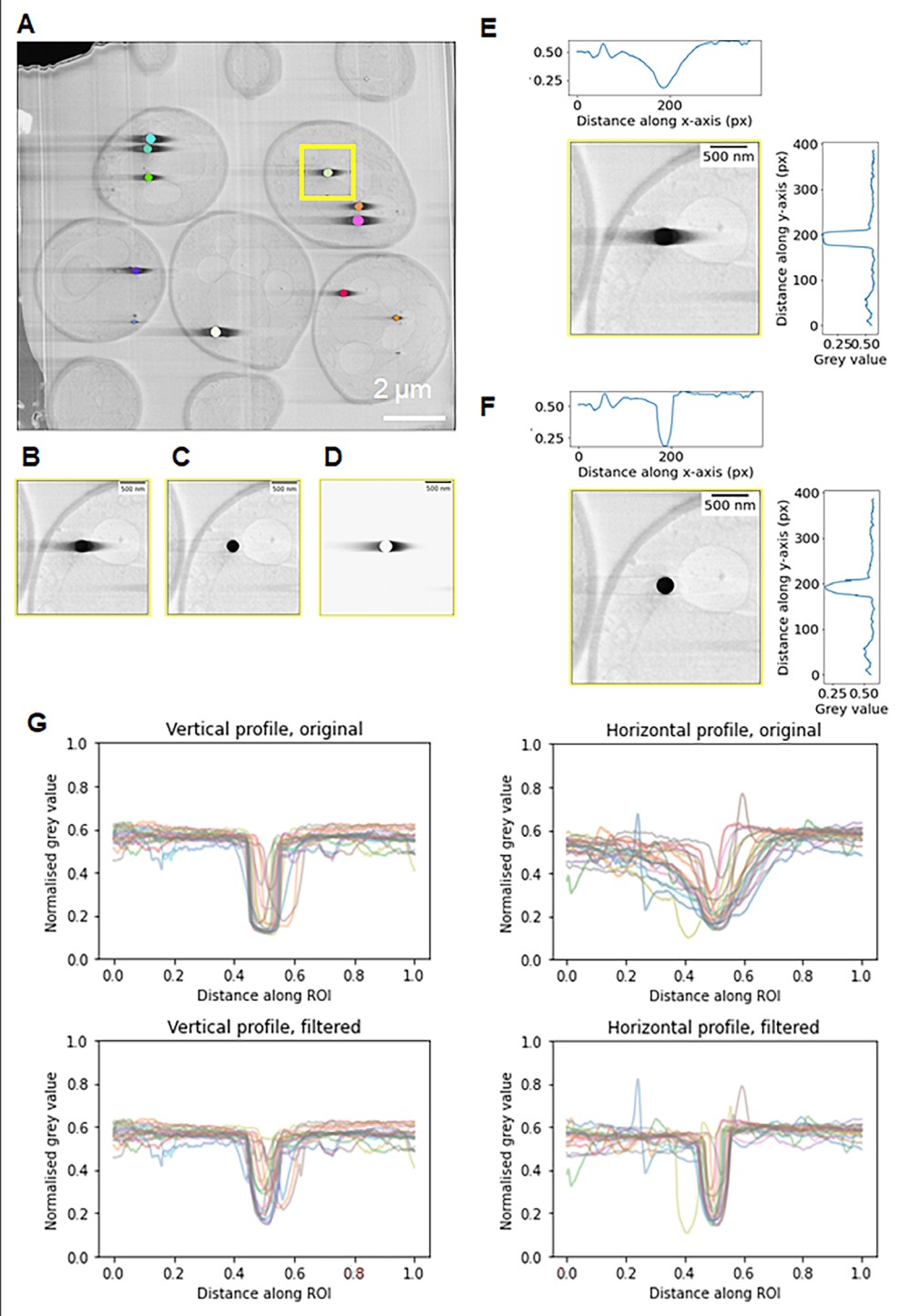

**Figure 10.** Post-processing computational approach to mitigate charging artefacts. Cells were imaged normal (90°) to the surface and argon used for milling. (**A**) Depiction of the results of instance segmentation of lipid droplets found within the serial plasma focussed ion beam (pFIB)/scanning electron microscopy (SEM) volume. The lipid droplet highlighted in the yellow box in (**A**) is shown inset (**B**) with the same region shown in (**C**) after charging artefact suppression. The inset (**D**) shows the charging artefacts in isolation and demonstrates their asymmetry – this is produced when (**C**) is subtracted from (**B**). Since the artefact is asymmetric, different functions were fitted on the left and right of the charging artefacts. (**E**) and (**F**) are vertical and horizontal grey value line profiles through the lipid droplet in (**B**) and (**C**), respectively. (**G**) aggregated vertical and horizontal line profiles for 20 lipid droplets from (**A**) show that filtering to remove charging artefacts restores the sharp dip in grey values for the lipid droplets

*Figure 10 continued on next page*

*Figure 10 continued*

in the horizontal line profile, and more closely matches the grey values on either side of the charging centre, as can be seen in the vertical line profiles (*Pennington et al., 2022*).

## Machine learning aided segmentation

In volumetric imaging, the segmentation of biological features for annotation and analysis remains a bottleneck due to the laborious and time-consuming nature of manual annotation. Automation of this process by utilising machine learning tools can greatly reduce the time taken to create a segmentation. Here, we demonstrate the automated segmentation of mitochondria within a cryo-serial pFIB/SEM dataset of heart tissue using the neural network trained model generated in SuRVos2 and evaluate its accuracy (*Figure 11*).

By providing an initial annotation identifying the mitochondria, cytoplasm, and other features such as red blood cells, a U-Net++ model was trained and predicted segmentation across the whole volume (*Figure 11B*). A small region of the volume (*Figure 11A and B*, red box) was annotated manually to establish a 'ground truth' in which to evaluate the performance of the automated segmentation. By comparison with a manually segmented region, we could show that automated segmentation largely agrees with the manual segmentation in identifying mitochondria with a Dice score of 0.87 (*Figure 11C–E*). Additionally, the percentage of the smaller ROI segmented by SuRVos2 (24%) was largely in agreement with the percentage segmented manually (20%). Moreover, 15 'components' were counted by SuRVos2 after the removal of segmentations with less than 1000 voxels, which is different from the total of 23 whole or part mitochondria counted in the manual segmentation. Additional work would be required to better quantify the errors on counts obtained from the automated

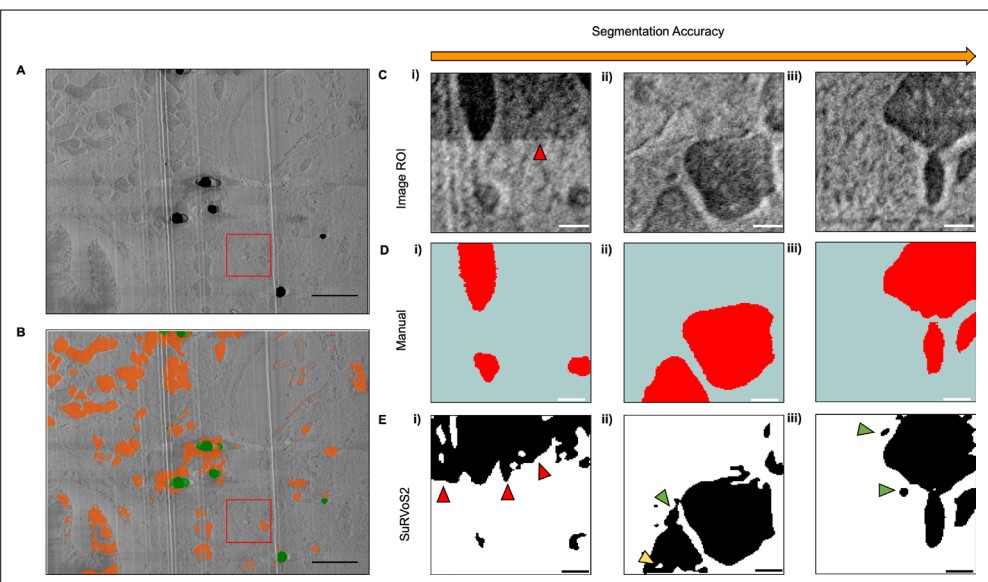

**Figure 11.** Automated segmentation of subcellular features using SuRVoS2. (**A**) A representative *XY* slice from the serial plasma focussed ion beam (pFIB)-scanning electron microscopy (SEM) of heart tissue data. (**B**) The same SEM image overlaid with the automated segmentation produced by SurVoS2 after training a multi-axis U-Net++ (orange). Artefacts such as lipid droplets are masked out of the segmentation (green). A section of the serial pFIB/SEM data was used to compare the manual segmentation of mitochondria vs. SuRVoS2 segmentation (red box). For A and B, scale bar = 1.2 µm. (**C**) Evaluation of the automated segmentation output from SuRVoS2. A 128 × 128 × 128 region within the top right quadrant of the serial pFIB/SEM data was extracted (C i–iii) and the mitochondria within *XY* slices were segmented manually to establish a 'ground truth' (D i–iii). The trained multi-axis U-Net++ network then automatically segmented the region (E i–iii). The Dice score between the segmentation and ground truth is 0.87, with the qualitative performance of the segmentation increasing in accuracy from left to right. In the less accurate automated segmentations, changes in contrast due to charging were erroneously segmented (left column, red arrows). Small regions within mitochondria are not segmented (E ii, yellow arrow) whilst some small regions that are not part of a mitochondria are segmented incorrectly (E ii–iii, green arrows). Scale bar: 250 nm.

segmentation, however these results set a baseline that demonstrates cryo-serial pFIB/SEM data can be segmented in an automated fashion.

## Discussion

Imaging of biological materials using SEM is well known but challenging since biological material is primarily composed of light atoms (C, N, O) in a water background. This is overcome by fixation and heavy metal staining which generates contrast (*Knott et al., 2008*; *Watson, 1958*). This is a very powerful approach and can be repeated using some form of ablation (FIB, knife) to expose fresh surface in a serial process that can be parallelised enabling large volumes to be imaged (*Hughes et al., 2014*; *Starborg and Kadler, 2015*). However, the process of fixation and staining by their nature alter the underlying biological structures (*Thompson et al., 2016*). Furthermore, these treatments are incompatible with further downstream analysis by high-resolution cryo-ET.

Building on work which applied SEM on vitrified hydrated samples (*Schertel et al., 2013*), we have evaluated different plasma gases for their ability to remove bulk biological materials while forming a smooth surface with minimal curtaining artefacts. Our work demonstrates that all four gases tested (O, N, Ar, and Xe) (*Figure 1*) can under some conditions produce smooth surfaces that are amenable to SEM imaging. The precise behaviour of the different plasma species differs depending on the ion current used (*Figure 1*) and for nitrogen on the acceleration voltage used (*Figure 1—figure supplement 2*). Our results show that there is a difference in the curtaining propensity depending on the used gases and ion current, with N and O showing large curtaining propensity whilst Xe and Ar showed lower propensity. We speculate that monatomic gases are less reactive at the surface of biological material, and less prone to curtaining.

pFIB milling is effective for both plunge-frozen cells and HPF mouse tissue. We observed a higher tendency to produce curtain artefacts in brain and heart tissue, something we attribute to the greater initial unevenness of the material due to the presence of lumen, facias, and supportive layers which mill at different rates. We used gaseous organoplatinum to coat and stabilise the sample. Since both composition and application method of the protective layer alter curtaining propensity (*Leer et al., 2009*), there may be gains from further exploration of different coating approaches.

The curtaining score that we have developed does provide near real-time feedback on the extent of curtaining. In future, a more refined version of this measure could be used on-the-fly to automatically adjust milling parameters to prevent (and eliminate) curtaining during data acquisition.

The use of diatomic gasses (O, N) to generate the plasma led to double images which, as we showed, can be (to an extent) corrected for but required more manual operation (*Figure 6*) as there was a high failure rate of automation (>80% vs. <10% for argon). The use of oxygen plasma on resin-embedded and stained samples has been reported to be effective (*Gorelick et al., 2018*) but a success rate was not reported. Since our goal was to develop automated procedures for vitrified samples, we did not pursue oxygen or nitrogen gases further. Ar and Xe have similarly low curtaining propensity at low current; thus, both are strong candidates for exploitation in this workflow. At high currents Ar performed better, as demonstrated by its reduced curtaining score and visual lack of curtains. We focussed argon to demonstrate the potential of our approach.

The use of pFIB to fabricate lamella has been demonstrated with plunge-frozen vitrified materials as part of a high-resolution high-throughput cryo-ET workflow (*Berger et al., 2023*). The cryo-ET workflow study indicated that milling with Ar plasma (30 kV, 60 pA) beam adversely affected the resolution of sub-tomographically averaged ribosomes within around 45 nm of the surface. This effect was present despite the 'smooth' surface of the lamella and was attributed to damage introduced by argon ions colliding with and penetrating the sample. Thus, while a lack of curtaining is a necessary condition for SEM imaging, it does not necessarily indicate a lack of damage. For serial pFIB/SEM, the Wattage used for argon is twofold higher than that used for the final step of lamella polishing (200 pA at 20 kV vs. 50 pA at 30 kV). Intuitively, we would expect damage (not just curtaining) to increase with increasing current, although this has not been established. When we compared cryo-serial pFIB/SEM (low current) images of Vero (milled with nitrogen plasma) and HeLa (argon plasma) cells, nitrogen showed higher resolution. It is possible that damage may be dependent on the plasma not simply on its current and such damage may not simply correlate with curtaining, but further investigation is required. SEM imaging itself will damage the sample; the greatest dose we employed during data acquisition was under $10^{-2}$ e$^-$/Å$^2$ over the whole scanned image, even with the smallest pixel sizes used

for imaging (*Table 1*). This is a lower dose than those used during cryo-ET imaging (*Berger et al., 2023*). SEM damage would be expected to be confined to the interaction depth, less than 50 nm, and therefore, we would expect that SEM-induced damage would be contained within the same volume (close to the surface) as the FIB damage, which is approximately 45 nm (*Berger et al., 2023*). However, further experimentation may disclose unexpected effects.

Using low current, we were able to mill in a highly controlled manner, accurately ablating a 50 nm layer reliably. Importantly, the absolute size of the beam spot is not necessarily the limiting factor for milling resolution – fine control of the beam (homogeneity of beam current), careful placement of milling boxes, and a steep beam profile can enable slice thickness or final lamella thickness smaller than the beam diameter. Therefore, for plasma ion beams we expect it will be possible to ablate thinner layers. Since SEM images are projections which lack depth information the ideal milling step will match the depth of penetration (electron interaction) of the SEM; estimated to be between 20 and 30 nm (*Kanaya and Okayama, 1972*). Moreover, the depth of penetration is a function of the elements involved and will vary by sample. In this study 50 nm was chosen as a compromise since smaller steps require more time and potentially introduce cumulative electron damage. We did not notice obvious vertical discontinuities due to excessive electron dose (SEM) and features were tracked across the FOV in serial images. The rate limiting step in the automated workflow was the SEM imaging (6–7 min), with 30 s on average taken for milling of a 20×4 µm rectangle using Ar. Thus, using a plasma which mills faster at low power currently would have no significant improvement in the overall time taken for cryo-serial pFIB/SEM. However, increasing the speed of SEM would be extremely beneficial. Scanning multiple lines concomitantly and/or software solutions (*de Haan et al., 2019*) would enable the analysis of larger volumes and would improve the integration of serial pFIB/SEM into other pipelines.

The quality of the resulting cryo-serial pFIB/SEM images was excellent even with sub-optimal SEM imaging geometry. Using an algorithm derived from fluorescent imaging we calculated the local resolution across the image. This varies between samples and data acquisition parameters. Most importantly, and not surprisingly, the pixel size is critical. In our experiments, patch size was maintained at 256 x 256 for all pixel sizes, which may lead to more 'optimistic' resolution estimates. Therefore, smaller patch sizes and robust local analyses will make this approach more robust. Nonetheless, our observation of macromolecular assemblies/features is consistent with the calculated resolution of better than 50 nm.

By imaging 500 slices of 50 nm depth from HeLa cells, we were able to identify macromolecular complexes, such as NPCs and centrioles. We found two centrosomes indicating a cell in prophase but did not observe the spindle microtubules. In Vero cells, we delineated the mitochondrial network in exquisite detail and were even able to quantify the ER to organelle contacts. In mouse brain tissue, we visualised complete synapses including pre-synaptic vesicles (approx. 30–50 nm), identified myelin sheaths, and cell-cell interactions. Previous studies (*Schertel et al., 2013*) using SEM on high-pressure frozen cryogenic samples have demonstrated the applicability of SEM to visualise subcellular structures, including the perinuclear space. Our approach was able to identify large macromolecular complexes even within tissues, representing a significant improvement in the power of the approach.

Serial pFIB/SEM images could be correlated with fluorescence and TEM images showing promise for a correlated workflow. For targeting-specific ROIs within cells or tissues, SEM imaging at lower resolutions may be possible and would increase throughput of targeting ROIs on the fly. We have yet to determine the consequence of the highest milling currents upon the quality of TEM imaging. We did not observe devitrification in the mammalian cells or bacteria we imaged. Poor vitrification was observed in the media surrounding the bacteria. Comparison to non-milled samples showed that the poor vitrification resulted from sample preparation (David Farmer, personal communication, 2022).

In all samples, we did observe sample charging. This was most severe on lipid-rich regions. We implemented a software mitigation approach that reduced the effect of charging on the surrounding data, though this was unable to recover information that was otherwise lost due to charging. Of course, preventing or reducing these artefacts during acquisition of data would be preferable.

Applying an automated, machine learning segmentation software to a serial pFIB/SEM data dataset resulted in a predicted segmentation with a Dice score of 0.87. This segmentation was largely in agreement with the manually segmented 'ground truth' but was less accurate in locations where artefacts such as charging (*Figure 11*) were present. The enhanced contrast from imaging the sample

**Table 1.** Serial plasma focussed ion beam (pFIB)/scanning electron microscope (SEM) parameters used for the different biological samples.

| Sample | Milling gas | Acceleration voltage (kV) | Target current (nA) | Software | SEM imaging angle (°) | Pixel size (nm) | SEM voltage (kV) | SEM current (pA) | Dwell time (ns) | Line integration | Gain (%) | Offset voltage (V) | Electron dose (e⁻/A²) |
|---|---|---|---|---|---|---|---|---|---|---|---|---|---|
| HeLa | Argon | 20 | 0.2 | TFS ASV | 52 | 6.745 | 1.25 | 6.25 | 100 | 100 | 62.9 | −12 | $8.24 \times 10^{-4}$ |
| Vero | Nitrogen | 30 | 0.27 | Serial FIB | 90 | 1.927 | 1.2 | 6.25 | 100 | 100 | 56.45 | −8.4 | $9.84 \times 10^{-3}$ |
| Rubrum | Argon | 20 | 0.2 | Serial FIB | 90 | 1.942 | 1.1 | 6.25 | 100 | 100 | 36 | −3.37 | $9.94 \times 10^{-3}$ |
| Brain | Argon | 20 | 0.2 | TFS ASV | 52 | 6.745 | 1.1 | 6.25 | 500 | 16 | 40.25 | −4.37 | $8.24 \times 10^{-4}$ |
| Heart | Argon | 20 | 0.2 | TFS ASV | 90 | 5.62 | 1.1 | 6.25 | 100 | 100 | 45.42 | −4.41 | $1.19 \times 10^{-4}$ |

Formula from current measured using Faraday cup.

perpendicular to the SEM paired with the post-processing charge removal allowed for volumes to be segmented automatically. Serial pFIB/SEM can thus be utilised to provide quantitative information on the subcellular organisation within tissues.

A systematic comparison of *Chlamydia*-infected cells showed clear improvements in the definition of the ultrastructure imaging perpendicular to the surface. This was most clearly seen in the sharpness of the bacterial inner and outer membranes with the infected cells (*Figure 4*). The difference in quality between the imaging regimes was reduced but not eliminated by careful manual optimisation of the working distance (*Figure 4*). The perpendicular imaging regime does require additional movements of the stage which for a highly repetitive process can lengthen the time taken and introduce compounding errors in stage position. However, the additional time was insignificant, and we demonstrated that these additional movements have a negligible effect on the stage. We were able to segment both mature and separately developing chromatophores in the photosynthetic bacteria, *R. rubrum*. We also imaged unstained (but fixed) mouse heart. We were able to measure the *Z*-disc to *Z*-disc distance as being ~1.68 µm, which could reflect the heart being in diastole, but more likely demonstrates that the sarcomeres had undergone contraction post-mortem where typical length in vivo is ~1.7–2.3 µm (*de Tombe and ter Keurs, 2016*). The images allowed accurate assessment of the mitochondrial density in this tissue and quantitative comparison with the brain tissue. The ability to visualise protein assemblies and ultrastructure in vitrified unstained tissue with laboratory-based equipment has potential applications in pathology, where early identification of disease mechanisms remains an unmet challenge.

Cryo-serial pFIB/SEM is a useful addition to the structural biology toolbox. We have shown that it can accurately determine biological ultrastructural arrangements from simple bacteria to complex tissue. The approach can generate a 3D map of biological materials. The approach has several key advantages: it is highly controllable, it is relatively rapid, can be done in a laboratory, and it is compatible with subsequent imaging modalities.

## Materials and methods
### Sample preparation
#### Mammalian cells

HeLa cells (CCL-2, ATCC, Manassas, VA, USA) were grown in high glucose DMEM with non-essential amino acids supplemented with 10% (v/v) foetal bovine serum (FBS; Gibco, Thermo Fisher Scientific, Waltham, MA, USA), 1% (v/v) penicillin/streptomycin (P/S; Gibco) 1% (v/v) glutamine (200 mM stock, Gibco). Vero cells (CCL-81, ATCC) were grown in DMEM (Gibco) with 10% FBS, 1% P/S, and 25 mM HEPES (Gibco). RPE-1 hTERT (CRL4000, ATCC) were grown in DMEM/F12 with 10% FBS and 1% P/S. Cells were grown in a laboratory which implements routine tests for mycoplasma and upon take up of a new cell line. Cells were maintained at 37°C and 5% $CO_2$ at >80% humidity. Cells were then seeded onto gold grids (either 'UltrAuFoil' 200 Au mesh, R2/2 gold film, or Quantifoil 300 Au mesh, R2/2 carbon film; Quantifoil Micro Tools, Großlöbichau, Germany) and grown to 50% confluency before plunge freezing. Grids were then plunge-frozen in nitrogen-cooled liquid ethane using a Vitrobot (Thermo Fisher Scientific). Grids were then clipped into Autogrids (Thermo Fisher Scientific) and stored in liquid nitrogen.

#### C. trachomatis

The intracellular strict gram-negative bacteria *C. trachomatis* was prepared as previously described (Dumoux, 2012). HeLa cells were cultured on grids and 24 hr after seeding infected with *C. trachomatis* LGV2 as previously described (Dumoux, 2012). 24 hr post infection cells were plunge-frozen using a Vitrobot (Thermo Fisher Scientific). Cells were pre-incubated on the grid with media containing 10% glycerol.

#### R. rubrum

*R. rubrum* (gift by D Canniffe from the University of Liverpool) were grown photosynthetically on modified Rhodospirillaceae (DSMZ 27) medium as previously described (*García-Sánchez et al., 2018*). Bacteria were exposed to 100 µmol photons/m²/s[1] illumination at 30°C in 150 ml flasks until an A680

of 1.6. 50 ml of culture was harvested by centrifugation at 3000 × *g* for 30 min then resuspended with 15 ml of 20 mM Tris buffer at pH 7 and used for grid preparation.

8 µl of sample was applied to a plasma cleaned (Harrick plasma cleaner, medium setting, 40 s) Quantifoil Cu 300 R2/2 grid and plunge-frozen in liquid ethane with a GP2 (Leica Microsystems, Wetzlar, Germany) blotted from behind, 6 s blotting time; 15°C and 75% chamber temperature and humidity, respectively.

## Brain tissue

68-day-old Protamine-EGFP (PRM1-EGFP) mice (*Haueter et al., 2010*) (CD1; B6D2-Tg [(Prm1-EGFP)]#Ltku/H) were euthanised in a schedule 1 procedure via intraperitoneal injection of sodium pentobarbital followed by decapitation following licensed procedures approved by the Mary Lyon Centre and the Home Office UK. Brains were dissected and cut into four equidistant lateral sections using a scalpel, with region 1 encompassing the olfactory bulb. Regions 2 and 3 were then further sectioned using a vibratome (VT1000S, Leica) set to produce 200 µm sections. Brain sections were kept at 4°C in Hank's Balanced Salt Solution (HBSS) from death to HPF. A 2 mm biopsy punch was used to excise regions of cortex from lateral slices.

Punches were then placed onto electron microscope grids during freezing. Brain punches on grids were frozen between 3 mm planchettes/carriers (Science Services, Munich, Germany) assembled on mid-plates. Briefly, planchettes were coated with 1% soya-lecithin dissolved in chloroform and the solution allowed to evaporate. This process generates small microvesicles on the surface of the planchette. A flat-sided planchette was placed flat side upwards and a glow discharged (Glocube, Quorum, Lewes, UK) electron microscopy grids (UltraAuFoil, 200 Au mesh, 2/2 Au film; Quantifoil Micro Tools) placed on top; brain punches were then placed onto the grid and submerged in 20% bovine serum albumin (BSA) in HBSS. 3 mm planchettes with a 0.1 or 0.2 mm recess, also coated with 1% soya lecithin, were then placed recess-side-down onto the assembled sandwich and high-pressure frozen in a Leica HPM 100 (Leica Microsystems). The planchette grids were then disassembled under liquid nitrogen, clipped into Autogrids (Thermo Fisher Scientific) and stored at 80 K (liquid nitrogen) for later use.

## Heart tissue

An 8-day-old 'wildtype' C56BL/J mouse was euthanised in a schedule 1 procedure via intraperitoneal injection of sodium pentobarbital followed by decapitation following licensed procedures approved by the Mary Lyon Centre and the Home Office UK as described for the brain tissue. Heart was dissected and placed in ice-cold Millonig's buffer (Fisher Scientific, Thermo Fisher Scientific) then fixed using ice-cold 4% PFA in Millonig's buffer (Fisher Scientific) then left to incubate overnight at 4°C. Thin sections of 50 µm were obtained using a vibratome and placed onto a glow discharged electron microscopy grid (UltrAuFoil, 200 Au mesh, 2/2 Au film; Quantifoil Micro Tools) that had been pre-clipped into Autogrids (Thermo Fisher Scientific). The sample was assembled in the mid-plate between two 6 mm planchettes incubated with hexadecane for 15 min. Twenty percent w/v BSA (Sigma-Aldrich, St Louis, MO, USA) in PBS buffer was used as a cryoprotectant and filler and the assembly was high-pressure frozen in a Leica HPM 100 (Leica Microsystems). The planchette-autogrid assembly was disassembled and stored under liquid nitrogen.

## Yeast

*S. cerevisiae* were grown to an OD of 0.8 and plunge-frozen as described (*Khavnekar et al., 2022*).

## Beads

PEG 300-coated 1 µm polystyrene particles (Abvigen, Newark, NJ, USA) were diluted to 4 mg/ml and mixed 1:1 with 3× diluted 6 nm BSA-Gold nanoparticles (Aurion, Wageingen, Netherlands). Three µl of the resulting mixture was spotted onto glow-discharged copper EM grids (Quantifoil 300 Au mesh, R2/2 carbon film; Quantifoil Micro Tools). The grids were then blotted for 2 or 4 s (blot force –10, humidity 70%) before plunge freezing by rapid immersion in liquid ethane using the Vitrobot (Thermo Fisher Scientific). Grids were subsequently clipped.

### Silicon

The silicon used for spot burns was monocrystalline silicon provided by Thermo Fisher Scientific as an alignment sample.

## FIB/SEM

The serial pFIB/SEM imaging was performed on a dual-beam FIB/SEM 'Helios G4 Hydra' equipped with an Aquilos II type cryo-stage, a four source 'Hydra' plasma ion column and an 'Elstar' SEM column (Thermo Fisher Scientific). The detectors are an Everhart-Thornley Detector and a through-the-lens detector using a positive bias (70 V) suction tube. An immersion field is generated to improve detection and signal to noise. The samples were loaded into a custom purpose 25° pre-tilt holder. During the experiments, an instrument defect which required the development of a new ion gun design affected plasma ion beam capability, preventing the usage of argon at 30 kV. All experimental dataset using argon as source for milling were performed at 20 kV. We were able to produce a complete set of curtaining scores for all gas at 20 and 30 kV after the source had been successfully replaced.

### Focused ion beam (FIB) milling

The accelerating voltage used for ion beam milling was 30 kV, except for argon which was used at 20 or 30 kV. To protect the leading edge while milling and to reduce surface topography, an organic platinum layer (trimethyl(methylcyclopentadienyl)platinum(IV)) of ~1–2 µm was deposited on the sample using a gas injection system (GIS). To reduce charging the sample was coated with two sputter-coated layers (before and after the organic platinum GIS layer) of platinum ions using an in-built platinum mass as a sputter target; removal of platinum from the mass deposits platinum vapour onto the sample. The mass is placed at the end of a small rod that can be inserted into the path of the ion beam (16 kV and ~1 µA, with full view of the rod). Thickness of the sputtered layer depends on the exposure time of the target to the ion beam. We usually deposit each layer for 45 s which gives a layer that is tens of nanometres in thickness. Once an area of interest was identified, the sample is placed at eucentric height, and a first opening trench is milled at ~2 nA. This enables a check that the sample of interest is in the milled area and assess the sample quality. The surface is then polished before starting the milling and imaging sequence.

### SEM

SEM image parameters are shown in *Table 1*. Image acquisition took between 2 and 5 min. Parameters for the dataset are as below.

The electron dose was calculated using this formula:

$$ED = \left(\frac{N}{A}\right) xt$$

where ED is the electron dose in $e^-/nm^2$, $N$ is the number of electrons, $A$ is the area, and $t$ the total time the sample was exposed. To calculate $N$, we used $N=I/e$, where $I$ is the current measured using a Faraday cup and $e$ is the charge of an electron.

## Fluorescence microscopy

A 'Meteor' fluorescent module (Delmic) equipped with a 50× (NA: 0.8) installed on our Helios G4 to streamline fluorescence imaging was used to perform fluorescence imaging. Without protective layers, an area without damages on support film and minimal ice contamination was selected as close to the centre as possible to ease subsequent TEM acquisition. Once the area identified we milled different marker on the grid bars and acquired a 3 by 3 fluorescence montage (excitation: 470/20 nm/ emission: 525/30 nm) of 20 µm thick z-series to compensate for uneven surface (Z step 500 nm).

### SerialFIB

The SerialFIB software (*Klumpe et al., 2021*) was run on the microscope via the AutoScript 4 (Thermo Fisher Scientific) interface. To extend SerialFIB to plasma ion sources, the template pattern files that follow the Thermo Fisher standard were adjusted to a pFIB equivalent to allow reading and writing of patterns. Parameters that changed in those files between a system utilising gallium, that is, an

Aquilos 2 dual-beam FIB/SEM (Thermo Fisher Scientific), to the system used in the current study utilising plasma-based milling were: total diameter, volume per dose, total beam area, sputter rates, and depth per pass. One crucial parameter that had to be adjusted when switch ion sources was the milling pattern pitch, as the pitch in plasma-based ion sources needs to be significantly higher than when milling with gallium-based liquid metal ion sources, likely due to the decrease in focussing capability of the pFIB. The pitch values were 9.5 nm for gallium and 116 nm for nitrogen and argon-based milling. Furthermore, the beam shift limits for the ion and electron column had to be adjusted to the limits on the Helios Hydra system.

The imaging script that allows for definition of a milling and another imaging position to allow for stage movements between the two steps in the volume imaging protocol was developed inside SerialFIB's scripting interface. In brief, the function allowing for serial FIB/SEM volume imaging runs was modified to take one-stage position for slicing and one-stage position for imaging into account. While stage movements normally lead to adjustment to the SEM focus according to linked stage height, the focus was enforced to a user-defined or via autofocussed determined value to remove the necessity of autofocussing after every stage movement. In addition, the horizontal field width was locked to avoid slight changes when readjusting focus after stage movements to avoid changes in the pixel size.

The script and all code for operating the microscope utilised here is available on the SerialFIB GitHub https://github.com/sklumpe/SerialFIB/ (copy archived at *Dumoux, 2023*).

## Auto Slice and View(ASV) (Thermo Fisher Scientific)

ASV version 4.2 was run on the Helios G4 Hydra. A fiducial was manually milled on the side of the target to assist the drift correction during milling. The fiducial should have a 'Uniqueness score' determined by the software of minimally 70%. No Y-correction was applied. On the preparation tab, only the 'Green clean pattern' was used to polish the surface (5–10 slices with an overlap of 5–10). Regarding the imaging parameters, no alignment, Y shift correction, autofocus, autosource tilt, autocontrast, and brightness was used, and we did not use tiling.

## Lamella preparation

Lamella were prepared after automated serial pFIB/SEM. Once the serial pFIB/SEM performed, the surface was then either left as is or polished using a 50 nm (Y height) milling pattern using a 30 kV, 60 pA beam. Subsequently, the bottom of the lamella was generated using a milling pattern that was placed 1 µm away from the lamella edge with a 7 µm Y height. This area was then milled at 2 nA (30 kV). A box 300 nm from the lamella edge with a 2 µm Y height was then used for milling at 0.74 nA, before final milling to 200 nm final lamella targeted thickness using 60 pA.

## TEM

Grids containing lamellae were transferred to a 300 kV Titan Krios (Thermo Fisher Scientific) equipped with a Falcon 4i and a Selectris energy filter. Search maps were acquired using patches of 2048×2048 (2× downsample) with a pixel size of 22 Å. The number and layout of the patch was determined for each lamella. Dose-symmetric tilt series were acquired using TOMO 5 (Thermo Fisher Scientific) with a pixel size of 1.98 Å with a range of 102° with a 2° increment.

## Image metrics

### Beam profile

The beam profile was measured using the spot burn method in monocrystalline silicon. Two spots were milled using the spot tool in the XTUI software on a Helios Hydra. The beam was then switched back to xenon, if needed, for selective area platinum deposition using the GIS. Three-hundred nm of platinum was deposited to cover the spots. This was then cross-sectioned using xenon to allow for the FWHM to be measured from the cross-sectional SEM image. The SEM images were taken in immersion mode at 5 kV, 25 pA, 200 ns dwell with line integration of 8 and image resolution of 3072×2048 pixels. The iSPI tool was used to take images sequentially through the cross-sectioning mill to ensure the centre of the spots was imaged. The FWHM was measured manually using ImageJ. Four spots were averaged to give a mean and standard deviation.

**Table 2.** Focussed ion beam (FIB) current (targeted value) in nA.

|  | Argon | Xenon | Nitrogen | Oxygen |
|---|---|---|---|---|
|  | 6.2 | 3.3 | 14 | 4.2 |
|  | 2.4 | 1 | 1.8 | 1.4 |
| 20 kV | 0.2 | 0.1 | 0.32 | 0.29 |
|  | 7.6 | 4 | 24 | 5.6 |
|  | 2 | 1 | 1.7 | 1.7 |
| 30 kV | 0.2 | 0.1 | 0.27 | 0.23 |

## Curtaining propensity

Plunged-frozen *C. trachomatis*-infected HeLa cells were coated with a protective organoplatinum layer and platinum sputter coated as described. Each plasma ion beam was aligned immediately prior to milling. To enable effective curtaining rate determination with nitrogen and oxygen, double spot compensation (*Figure 1—figure supplement 4*) was performed for each current.

As the practical usable current is determined by the apertures within the ion column, the current used for each gas for study is based on the use of the same aperture; for example, at the lowest aperture size used the currents for nitrogen, oxygen, xenon, and argon are 0.27 nA, 0.23 nA, and 0.1 and 0.2 nA, respectively. A list of currents for a given aperture is shown in *Table 2*. However, these values will change over time as the beam mills the aperture during usage.

A series of 5 windows of $2 \times 2.5 \times 2 \ \mu m^3$ were milled until the average grey level intensity for the window reached 10% of the initial signal. The acceleration voltage for the I-beam was kept at 30 or 20 kV. To accurately reflect the ion beam current applied on the sample, the current presented for this experiment is the measured current. The milled surface is then imaged 90° to the SEM without an electrostatic lens to avoid interaction of the magnetic field with the ion beam. This is crucial for imaging with oxygen and nitrogen, where the use of the pre-field immersion lens led to blurring of the ion beam images. The pixel size, voltage, current, electric gain, and voltage offset were kept identical. This was repeated three times per current (total of 15 windows per current).

As curtaining can vary significantly across a milled surface, a semi-automated, quantitative assessment of curtaining is necessary for un-biased comparison of ion sources and milling settings. We use a method based on the work from Hovden's group (*Schwartz et al., 2019*). Using Fiji (*Schindelin et al., 2012*) the initial image is subject to a fast Fourier transform (FFT). A horizontal 5° mask is applied and a reverse FFT performed. In this new image the vertical lines have been removed. By subtracting the initial and inverted masked FFT images, we can produce a mask to isolate those lines using a default threshold based on minimum cross entropy (*Li and Tam, 1998*). A merged image where this mask is superimposed to the initial image allows precise identification of the location where the milling was performed. We can then produce a histogram where the white pixels are part of the curtain. The percentage of this white pixels is calculated and forms the curtaining score (*Figure 1—figure supplement 1*).

## FRC calculation

The FRC calculation for measuring local resolution was derived from the method of *Koho et al., 2019*, which was originally developed for optical microscopy. Their one-image FRC calculation has a calibration factor applied, which was derived from optical microscopy images. This calibration aims to match the one-image FRC resolution value to the gold-standard two-image FRC value. To recalibrate this measure to SEM images, their calibration procedure was repeated (*Figure 4—figure supplement 1*).

Here, the one-image FRC was used to measure local resolution as a quantitative comparison between images at different acquisition settings and positions within the stack. The images to be studied here were split into patches of 256×256 pixels, and the image resolution for each patch was calculated with the newly calibrated one-image FRC measure.

To calibrate the one-image FRC for EM images, a calibration dataset was obtained which consisted of pairs of images taken of the same FOV at different pixel sizes (1.12, 2.25, and 4.5 nm/pixel). One set of images was taken at 52° SEM angle, and the other at 90°. Both images in each pair were registered

to each other with SIFT landmark placement using the linear stack alignment with SIFT plugin in Fiji 2.3.0/1.53q (*Schindelin et al., 2012*). The images were cropped to one FOV for all pixel sizes for each SEM angle, which covered 2000×2000 and 1600×1600 nm$^2$ for the 52° and 90° images, respectively, note that the number of pixels in each pair of images was different due to the varying pixel size.

The two-image FRC curves were calculated from each pair of images at each pixel size and SEM angle. Briefly, the cross-correlation between both images in the Fourier domain within frequency bands was determined, and the cross-correlation values were plotted against the frequency. The frequency was normalised against the maximum frequency values for each image to enable direct comparison between images. The normalised frequency at which the cross-correlation falls below 0.143 was determined, which is represented as $r_{ref}$ for the two-image FRC, and $r_{co1}$ for the uncalibrated one-image FRC. The one-image FRC curves were obtained for each of the images in the pair using the checkerboard sampling approach described in the work of Koho et al.

A calibration curve was fitted to the plot of $\frac{r_{co1}}{r_{ref}}$ vs. $r_{co1}$ (*Figure 4—figure supplement 1*). Application of this calibration factor to the one-image FRC curve shifted the curve to match the gold-standard two-image FRC measure, so the resolution value from the one-image FRC was comparable to the reference measurement (*Figure 4—figure supplement 1*). The original implementation of Koho et al. was modified to include this calibration so that it would be applicable to SEM images. This modified one-image FRC code is available as part of Quoll, an open-source Python package (*Ho, 2022*).

### Shift measurement

One μm PEG beads were used (see previous section). Argon 30 kV 200 pA was used. SEM images were acquired using 1.2 kV 6.25 pA exposing for 100 s. In order to reduce the charging artefact, we use line integration of 25.

To adequately calculate the extent to which the samples shift during tilting, SIFT landmark placement in Fiji/ImageJ 2.3.0/1.53q was used to automatically place matching landmarks between pairs of adjacent $Z$ slices. The Euclidean distance was calculated between each pair of landmarks with the formula:

$$d = \sqrt{(x_2 - x_1)^2 + (y_2 - y_1)^2}$$

The Euclidean distance between landmarks was divided by the diagonal length of the FOV of the image, to ensure a fair comparison between images of different sizes.

### Depth of field

A tin ball sample was used and, with the focus fixed, the stage height was adjusted in ~5 μm increments to obtain a through-focus series of images. The FRC method was applied to each of these to estimate the average resolution, giving a clear minimum (*Figure 4—figure supplement 4*, *Bäuerlein and Baumeister, 2021*). The depth of field was then taken as the distance through which no significant change in the image resolution was measured. This was carried out by a two-tailed, two-sample equal variance t-test with a confidence interval of 95% between the measured resolutions at each stage position with the stage position with the lowest measured resolutions. For both 1 and 2 kV, four-stage positions were found to be statistically similar (marked with asterisks in *Figure 4—figure supplement 4*), so the depth of field was therefore estimated to be ~20 μm.

### Sphericity (3D)

From the images obtained, an average size of the beads can be extracted and compared with manufacturer's values. Assuming the beads are perfectly spherical, the images are expected to show a cross-section of the bead as being circular. Mathematically, the area covered by the bead is expected to be consistent with the area of a circle with a varying radius along the perpendicular slice direction. This can be derived from the sphere equation, and the area is equal to:

$$A(z) = \left[ \left( \frac{d}{2} \right)^2 - \left( \alpha (z - z_c) \right)^2 \right]$$

with $d$ being the diameter of the bead, $z$ the slice coordinate, and $z_c$ the height centre position of the bead. The parameter $\alpha$ was added here to consider potential coefficient error in the milling thickness,

which in a perfectly calibrated milling-capable microscope would be exactly 1. In each image slice the beads were manually annotated in Napari, and the connected components tool in plugin pyclesperanto (*Sofroniew et al., 2022*) (https://github.com/clEsperanto/pyclesperanto_prototype) was used to collect the 2D area of each bead from the number of pixels, and plotted as a function of slice as dots, for each of the methods used. The 2D area of each bead was collected from the number of pixels, and plotted as a function of slice as dots, for each of the methods used. The respective lines represent optimised $A\left(z\right)$ (https://docs.scipy.org/doc/scipy/reference/generated/scipy.optimize.curve_fit.html) curves, with parameters $d$, $z_c$, and $\alpha$ being allowed to vary for each of the beads independently.

## Circularity (2D) (*Chlamydia*-infected cells)

Bacteria were manually segmented, and the outline analysed using the 'Analyze Particles' tools from FIJI (*Schindelin et al., 2012*). The average and standard deviation were then calculated and shown in *Figure 4*.

## Charging artefact mitigation

Segmentation of charging centres was completed using a FastAI version 1.06 (*Howard and Gugger, 2020*) implementation of a 2D U-Net (*Ronneberger et al., 2015*). The U-Net was initially set up with an encoder based in resnet34 and training was done on 256 pixel × 256 pixel patches of 1024 test images and respective manually segmented labels, with and without artefacts at a ratio of 1:1. Then, this data was split into training (80%) and validation (20%) data. The preprocessing transforms used were FastAI's package defaults and with imagenet normalisation. The loss function used was binary cross entropy (bce) applied to the logits, and weight decay was set initially to 0.01. A total of 10 epochs were run resulting in IoU score in the validation data of 0.90. This model was then used to automatically segment the charging artefacts of the whole data volume. Manual correction of this segmentation was undertaken and then compared using Dice coefficient (*Dice, 1945*) to the automated segmentation. The segmented and manually adjusted dataset was used for subsequent work. After segmentation, local areas near charging centres were assessed on a row-by-row basis, both from the top and bottom of the image. Areas with charging artefacts were then filtered using the average of the previous 20 rows. The following smoothly varying continuous functions were then fitted and subtracted from the filtered data line (https://github.com/rosalindfranklininstitute/chafer) (*Perdigao, 2023*):

$$f_{left} = A\left(\frac{1}{e^{\left(\frac{x-x_0}{\sigma}\right)}+1} - 1\right), f_{right} = A\left(\frac{-1}{e^{\left(\frac{x-x_0}{\sigma}\right)}+1}\right),$$

with $A$, $x_0$, and $\sigma$ being parameters to be fitted, and functions $f$ fitted on the left of the labelled artefact, and $f$ fitted on the right side. Other smoothing functions (gauss tail, exponential, 1/*x*) were tested but these gave far better overall results both in fitting to the shape of the horizontal tails and algorithmically successful in finding optimal solutions.

In total, 117 charge centres were identified during segmentation of the *S. cerevisiae* dataset. At each of these charging centres, an average line profile of the grey values in both the *X* and *Y* directions was taken. For the horizontal line profiles, the box over which the line profiles were averaged was 1× the charging centre diameter in height and 9× in width, while for the vertical line profiles, it was 1× in width and 9× in height. The box size was chosen empirically to capture the full extent of the charging artefact. Due to the presence of other charge centres nearby or to the edges of the image, 20 charging centres were assessed fully.

## Videos

All videos were aligned using SIFT (*Lowe, 2004*) and filtered using a mean (2 pixels) filter using either Fiji/ImageJ 2.3.0/1.53q or Amira 2020.2. The data acquisition parameters of the SEM are described in the FIB/SEM section.

## Manual segmentation and quantification

Amira Version 2020.2 (Thermo Fisher Scientific) was used to align and segment the dataset. The alignment was performed using the DualBeamWizard and segmentation was carried out manually. For volume/surface rendering, the object corresponding to each segmentation was exported as tiff stacks and visualised in 3D viewer in Fiji (*Schindelin et al., 2012*). For the quantification, the object containing the manually segmented boundary was exported from Amira as tiff stacks. These stacks were converted to binary 3D masks and ImageJ 3D object counter was run on the ROI. Results exported as a .csv document for analysis in Excel (Microsoft).

## Semi-automated segmentation

Automated segmentation of mitochondria within the serial pFIB/SEM dataset of heart tissue was performed using the SuRVos2 workbench (*Pennington et al., 2022*). Manual annotation of two ROIs containing mitochondria from two different areas of the dataset was carried out. The first ROI contained a representative morphology and contrast of mitochondria (dimension 136 × 1058 × 1123) and the second contained a less representative appearance of mitochondria (dimensions 136 × 582 × 611). For the U-Net training, three labels were used to label cytoplasm, organelle, and charging artefact. In total, 21% of the original volume was annotated. However, an initial crop removed an unusable portion of the image around the edges before annotation and segmentation. Consequently, the total amount of the usable data that was annotated for training the U-Net was 31% of the cropped area. After the U-Net prediction, manual annotation was added (time <1 hr) to separate out large features such as the red blood cells and the GIS front to produce the final segmentation.

The architecture chosen for segmentation was a multi-axis U-Net++. This trains in three separate axes but here only one axis was used for prediction (the z-axis). The U-Net++ architecture was preferred over the standard U-Net as the U-Net++ is often able to achieve a higher Dice score when the patterns to be learned are subtle or from lower contrast images. The model was trained using Dice loss for nine cycles with the weights of internal ResNet-34 of the encoder side of the U-Net frozen and six cycles with it unfrozen.

For evaluation of the predicted SuRVos2 segmentation, a region that had not been used in the training of the model (dimensions 128 × 128 × 128) was manually segmented using Amira Version 2020.2 (Thermo Fisher Scientific). All mitochondria within all slices were annotated for comparison. The evaluation of the predicted segmentation was done by comparing this 128 × 128 × 128 voxel 'ground truth' region of manual segmentation (Amira). The percentage of volume segmented for both the ground truth and the predicted segmentation was determined manually. The Dice score measurement between the segmentation and ground truth and the filtering of components was also performed using SuRVos2.

## Correlative microscopy – image alignment and stitching

The tile set generated by the TEM during the imaging of the lamella ('Search map') was individually renamed prior stitching using the Fiji plugin 'Stitching' using the 'Grid/Collection' pattern (*Preibisch et al., 2009*).

The images from the different microscopic techniques were aligned using the Fiji plugin 'BigWarp' (*Bogovic et al., 2016*).

## Acknowledgements

The authors would like to thank the experimental coordinators at Diamond Light Source for their diligence in supporting our instruments, including during the overnight runs. We would like to thank Chelsea Norman for her support in sample preparation, and Chloe Cheng for helpful discussion and feedback on the image processing workflows. We would like to thank Silvia da Graça Ramos and Despoina Eugenia Kiousi for their help with segmentation. We would like to thank Daniel Canniffe from the University of Liverpool for providing *R. rubrum* samples. We would like to thank Marianne Yon, Alexandra Rodrigues, and Sara Wells of the Mary Lyon Centre at MRC Harwell for their support in animal work. We would like to thank Jonathan Barnard for helpful discussions. The Rosalind Franklin Institute is funded by UK Research and Innovation through the Engineering and Physical Sciences

Research Council. Funding was also provided by the Wellcome Trust through the Electrifying Life Science grant (220526/Z/20/Z to JHN) and a Sir Henry Dale Fellowship (218579/Z/19/Z to LW).

## Additional information

### Competing interests
Ron Kelley: is an employee of ThermoFisher Scientific. Michele C Darrow: is an employee of SPT Labtech. The other authors declare that no competing interests exist.

### Funding

| Funder | Grant reference number | Author |
| --- | --- | --- |
| Wellcome Trust | 220526/Z/20/Z | James H Naismith |
| Wellcome Trust | 218579/Z/19/Z | Liang Wu |

The funders had no role in study design, data collection and interpretation, or the decision to submit the work for publication. For the purpose of Open Access, the authors have applied a CC BY public copyright license to any Author Accepted Manuscript version arising from this submission.

### Author contributions
Maud Dumoux, Conceptualization, Resources, Data curation, Software, Formal analysis, Supervision, Validation, Investigation, Visualization, Methodology, Writing – original draft, Project administration, Writing – review and editing; Thomas Glen, Resources, Data curation, Formal analysis, Validation, Investigation, Visualization, Methodology, Writing – original draft; Jake LR Smith, Data curation, Investigation, Visualization, Methodology, Writing – review and editing; Elaine ML Ho, Resources, Software, Validation, Investigation, Methodology, Writing – original draft, Writing – review and editing; Luis MA Perdigão, Software, Formal analysis, Validation, Investigation, Writing – original draft, Writing – review and editing; Avery Pennington, Software, Visualization, Methodology, Writing – review and editing; Sven Klumpe, Neville BY Yee, Software, Validation, Methodology, Writing – original draft, Writing – review and editing; David Andrew Farmer, Data curation, Methodology, Writing – original draft, Writing – review and editing; Pui YA Lai, William Bowles, Ron Kelley, Investigation, Methodology, Writing – review and editing; Jürgen M Plitzko, Software, Supervision, Methodology; Liang Wu, Investigation, Methodology; Mark Basham, Software, Supervision, Writing – review and editing; Daniel K Clare, C Alistair Siebert, Supervision, Investigation, Methodology, Writing – review and editing; Michele C Darrow, Software, Supervision, Investigation, Methodology, Writing – original draft, Writing – review and editing; James H Naismith, Conceptualization, Data curation, Supervision, Funding acquisition, Investigation, Methodology, Writing – original draft, Project administration, Writing – review and editing; Michael Grange, Conceptualization, Resources, Data curation, Software, Formal analysis, Supervision, Funding acquisition, Validation, Investigation, Visualization, Methodology, Writing – original draft, Project administration, Writing – review and editing

### Author ORCIDs
Maud Dumoux http://orcid.org/0000-0002-1732-1041
Neville BY Yee http://orcid.org/0000-0003-0349-3958
David Andrew Farmer http://orcid.org/0000-0001-5331-3551
William Bowles http://orcid.org/0000-0001-8115-4404
Jürgen M Plitzko http://orcid.org/0000-0002-6402-8315
Mark Basham http://orcid.org/0000-0002-8438-1415
Michele C Darrow http://orcid.org/0000-0001-6259-1684
Michael Grange http://orcid.org/0000-0003-2580-2299

### Ethics
All of the animals were handled according to approved and reviewed institutional animal care procedures. Mice were euthanised in a schedule 1 procedure via intraperitoneal injection of sodium pentobarbital followed by decapitation following licensed procedures approved by the Mary Lyon Centre

and the Home Office UK. All operating procedures were designed to minimise any suffering for the animals involved in the study.

## Decision letter and Author response
Decision letter https://doi.org/10.7554/eLife.83623.sa1
Author response https://doi.org/10.7554/eLife.83623.sa2

## Additional files

### Supplementary files
• MDAR checklist

### Data availability
Raw data, along with segmentation and associated TEM overviews (if relevant) are deposited on the EMPIAR data repository: EMPIAR-11414, EMPIAR-11415, EMPIAR-11416, EMPIAR-11417, EMPIAR-11418, EMPIAR-11419, EMPIAR-11420, EMPIAR-11421. Code can be found on the Rosalind Franklin Institute GitHub (https://github.com/rosalindfranklininstitute/chafer, copy archived at *Perdigao, 2023*; https://github.com/rosalindfranklininstitute/quoll, copy archived at *Ho, 2022*) and the serialFIB GitHub (https://github.com/sklumpe/SerialFIB/, copy archived at *Dumoux, 2023*).

The following datasets were generated:

| Author(s) | Year | Dataset title | Dataset URL | Database and Identifier |
|---|---|---|---|---|
| Dumoux M, Glen T, Smith JLR, Ho EML, Perdigão LMA, Pennington A, Klumpe S, Yee NBy, Farmer D, Bowles W, Kelley Ron, Plitzko JM, Wu L, Basham M, Clare DK, Alistair Siebert C, Darrow MC, Grange M, Yiu Pui, Audrey L | 2023 | Serial pFIB/SEM of PEG beads (test sample) | https://www.ebi.ac.uk/empiar/EMPIAR-11414/ | Electron Microscopy Public Image Archive, EMPIAR-11414 |
| Dumoux M, Glen T, Smith JLR, Ho EML, Perdigão LMA, Pennington A, Klumpe S, Yee NBy, Farmer D, Bowles W, Kelley Ron, Plitzko JM, Wu L, Basham M, Clare DK, Alistair Siebert C, Darrow MC, Naismith JH, Grange M, Pui Y, Audrey L | 2023 | Cryo serial FIB/SEM of mouse brain tissue | https://www.ebi.ac.uk/empiar/EMPIAR-11415/ | Electron Microscopy Public Image Archive, EMPIAR-11415 |
| Dumoux M, Glen T, Smith JLR, Ho EML, Perdigão LM A, Pennington A, Klumpe S, Yee NBy, Farmer D, Bowles W, Kelley Ron, Plitzko JM, Wu L, Basham M, Clare DK, Alistair Siebert C, Darrow MC, Naismith JH, Grange Ml, Yiu Pui, Audrey L | 2023 | Cryo serial FIB/SEM of mouse heart tissue | https://www.ebi.ac.uk/empiar/EMPIAR-11420/ | Electron Microscopy Public Image Archive, EMPIAR-11420 |

*Continued on next page*

*Continued*

| Author(s) | Year | Dataset title | Dataset URL | Database and Identifier |
|---|---|---|---|---|
| Dumoux M, Glen T, Smith JLR, Ho EML, Perdigão LMA, Pennington A, Klumpe S, Yee NBy, Farmer D, Bowles W, Kelley Ron, Plitzko JM, Wu L, Basham M, Clare D K, Alistair Siebert C, Darrow MC, Naismith JH, Grange M, Yiu Pui, Audrey L | 2023 | Cryo serial FIB/SEM of HeLa cells | https://www.ebi.ac.uk/empiar/EMPIAR-11419/ | Electron Microscopy Public Image Archive, EMPIAR-11419 |
| Dumoux M, Glen T, Smith JLR, Ho EML, Perdigão LMA, Pennington A, Klumpe S, Yee NBy, Farmer D, Bowles W, Kelley Ron, Plitzko JM, Wu L, Basham M, Clare DK, Alistair Siebert C, Darrow MC, Naismith JH, Grange M, Yiu Pui, Audrey L | 2023 | Cryo serial FIB/SEM of Rhodospirillum rubrum | https://www.ebi.ac.uk/empiar/EMPIAR-11418/ | Electron Microscopy Public Image Archive, EMPIAR-11418 |
| Dumoux M, Glen T, Smith JLR, Ho EML, Perdigão LMA, Pennington A, Yee NBy, Farmer D, Bowles W, Kelley Ron, Plitzko JM, Wu L, Basham M, Clare DK, Alistair Siebert C, Darrow MC, Naismith JH, Grange Ml, Yiu Pui, Audrey L | 2023 | Cryo serial FIB/SEM of RPE-1 cells | https://www.ebi.ac.uk/empiar/EMPIAR-11421/ | Electron Microscopy Public Image Archive, EMPIAR-11421 |
| Dumoux M, Glen T, Smith JLR, Ho EML, Perdigão LMA, Pennington A, Klumpe S, Yee NBy, Farmer D, Bowles W, Kelley Ron, Plitzko JM, Wu L, Basham M, Clare DK, Alistair Siebert C, Darrow MC, Naismith JH, Grange M, Yiu Pui, Audrey L | 2023 | Cryo serial FIB/SEM of Vero cells | https://www.ebi.ac.uk/empiar/EMPIAR-11417/ | Electron Microscopy Public Image Archive, EMPIAR-11417 |
| Dumoux M, Glen T, Smith JLR, Ho EML, Perdigão LMA, Pennington A, Klumpe Sven, Yee NBy, Farmer D, Bowles W, Kelley Ron, Plitzko JM, Wu L, Basham M, Clare DK, Alistair Siebert C, Darrow MC, Naismith JH, Grange M, Yiu Pui, Audrey L | 2023 | Cryo serial FIB/SEM of *Saccharomyces cerevisiae* | https://www.ebi.ac.uk/empiar/EMPIAR-11416/ | Electron Microscopy Public Image Archive, EMPIAR-11416 |

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
