## [Editor Report]

This important work is of interest to the electron microscopy and cell biology communities. The field has long searched for a suitable method to combine the pristine preservation of vitrified samples with a volumetric imaging modality that reveals subcellular architecture at sufficient contrast for ultrastructural analyses. The authors describe here the use of novel ion beams for imaging cellular samples in three dimensions, concluding that one of the four plasma sources tested produces the highest quality images, allowing them to provide several recommendations for imaging conditions along with software for improving collected images. This approach should be very useful for addressing many biological questions.

---

## [Decision Letter]

**Decision letter after peer review:**

Thank you for submitting your article "Cryo-plasma FIB/SEM volume imaging of biological specimens" for consideration by *eLife*. Your article has been reviewed by 3 peer reviewers, and the evaluation has been overseen by a Reviewing Editor and Anna Akhmanova as the Senior Editor. The following individuals involved in the review of your submission have agreed to reveal their identity: Alex de Marco (Reviewer #1); Alex J Noble (Reviewer #3).

Essential revisions:

1) In its current form the paper does not show or discuss how plasma beams can mitigate charging artifacts (or not). The authors should add a comparison of charging artifacts with different sources (including Gallium) before computational correction. If this is not possible, they should remove the claim that they 'address these challenges using a cryogenic plasma FIB/SEM' (including charging). Similarly, the resolution measurements presented in the last supplementary figures do not relate to the use of plasma FIB/SEM in comparison with other beams.

2) Add a description of curtaining over time, in response to the following point from reviewer 1: "curtains appear over time when milling, and it would be useful to understand how different sources behave over time in FIB/SEM tomography sessions. The score is currently done from individual windows milled, which gives a good indication of the performance. However, it would make sense to check that the behaviour remains identical in an imaging setting and with the moving milling windows (or lines). This will show the counteracting effect to the redeposition and etching effect reported when imaging with the E-beam the milled face. "

3) Add data on milling resolution and beam cross-section in response to the following point from reviewer 1: "No detail about the milling resolution has been reported. Since different currents and beams have different cross-sections, it is expected to affect the z-resolution achievable during an imaging session. It would be useful to have a description of the beam cross-sections at the various conditions used and how or whether these interfere with the preparation."

4) Comment on the effects of imaging tilt angle on collection efficiency.

5) Explain the loss in resolution upon tilting the imaging angle, and the effect on contrast on the curtain edges (refer to reviewer 1 comment: "the generation of secondary electrons is known to increase with the increased tilt and to consider that the curtains (that are the prominent feature on the surface) are running along the tilt direction, it would be expected to see no contrast difference between the background and the edge of each curtain as the generation of secondary electrons will increase with tilt for both the edges and the background. Therefore, the contrast should be invariant, at least on the curtains").

6) Comment on reviewer 1 claim of suboptimal astigmatism correction, and on how this might affect the comparison between the different imaging tilt angles.

7) Discuss whether and how the contrast gained from the optimised approach using plasma FIB will allow interpretation and segmentation of biological features.

Can automated segmentation approaches be applied to the FIB/SEM volumes in similar ways to cryo-ET reconstructions?

Could the authors demonstrate that the features detected in the SEM can be used to target areas of interest for cryo-ET? If so, can they show structural preservation in the cryo-ET reconstructions?

Alternatively, the manuscript should more explicitly state that downstream correlative approaches are a potential application to be developed and tested in future work, and this is currently not possible.

*Reviewer #2 (Recommendations for the authors):*

Suggestions for Improvement:

The results presented in this manuscript are substantial on their own, and merit publication. However, as written, I worry that most readers will have a similar experience as me, reading excitedly, on the edge of our seats, waiting for the proof-of-concept that this approach is effective (and perhaps more effective than CLEM) for targeting and downstream cryo-ET imaging, and will come up somewhat disappointed that this is not demonstrated.

I propose that the authors choose one of two possible paths to address the weakness above. The first, more extensive albeit thorough option, would be to address this by collecting tilt series data and reconstructing tomograms of the same lamella imaged by plasma cryo-FIB/SEM, and demonstrate that cross-correlation between volumetric cryo-FIB/SEM and cryo-ET acquisition is possible without affecting image/reconstruction quality. This could be evaluated either qualitatively (e.g. integrity of membranes/filaments) or structurally (e.g., by solving the subtomogram average of the cellular ribosome). Ideally, the authors would select a target and demonstrate an advantage of cryo-FIB/SEM targeting over the more standardized cryo-fluorescence/CLEM approaches.

The second option would be to revise the manuscript text to more explicitly state upfront that this work is solely focused on evaluating the performance of cryo-FIB/SEM for volumetric imaging and ultrastructural analyses, and that future work will determine whether these are of sufficient quality to the image by cryo-ET. The authors sort of dance around the possibility that this approach could be used for downstream applications without sufficiently testing this, and therefore, I believe the intent of the article needs to be more explicitly addressed in the manuscript text. This would strengthen the merits of the article by allowing the reader to focus on the substantial improvements to contrast/imaging without feeling like something is "missing" at the end.

Finally, I suggest that the authors apply an automated segmentation algorithm to their volumetric data as a more unbiased and practical metric to evaluate the degree to which cryo-FIB/SEM imaging increases resolution, contrast, and interpretability of subcellular features at scale that would enable biological insight.

*Reviewer #3 (Recommendations for the authors):*

The authors present a method for using the exceptional SEM imaging capabilities of the Helios Hydra and new cryo-plasma FIB milling functionality on vitrified biological specimens to obtain contextual information on cells and tissues while potentially localizing areas of interest for subsequent lamellae generation. Significant results are presented where analyses of the cryo-pFIB performance are qualitatively and quantitatively displayed and exemplified, which is particularly useful for the field as pFIBs begin to be used worldwide. The authors show SEM optimization examples from bacterial cells, mammalian cells, and tissue along with features that may be expected in the resolution range of the instrument (10 nm – 10 µm). The authors clearly compare the four plasma sources, proceed with the best source partly based on a proposed curtaining score, then optimize imaging quality by tilting to 90 degrees relative to the SEM, analyze several samples, and introduce a U-net algorithm for cleaning up charging streaks.

I have experience with cryo-FIB/SEM and cryo-ET/EM. Overall, I think this manuscript is timely and practical, given that cryo-pFIB/SEMs are anticipated to be adopted widely beginning in 2023, and should be published.

---

## [Author Response]

Essential revisions:1) In its current form the paper does not show or discuss how plasma beams can mitigate charging artifacts (or not). The authors should add a comparison of charging artifacts with different sources (including Gallium) before computational correction. If this is not possible, they should remove the claim that they 'address these challenges using a cryogenic plasma FIB/SEM' (including charging). Similarly, the resolution measurements presented in the last supplementary figures do not relate to the use of plasma FIB/SEM in comparison with other beams.

To address this comment, we clarified the role of the plasma, high resolution SEM and software in the abstract and have amended it; we did not mean to imply plasma mitigates charging. (Abstract Page 1, line 7 to 9). Indeed, to our knowledge, charging artefact is not a function of the ion beam source used for milling but is dependent on the energy applied (which we detail in the introduction; cross over – page 3, lines 1 to 7).

Our paper highlights a strategy to minimise the generation of charging by optimising the energy of the electrons used (a low kV column) and the imaging at 90 degrees relative to the sample plane.

The resolution metric presented in the manuscript assesses potential degradation of the SEM imaging quality during the serial pFIB/SEM process due to focussing issues, drifting, or accumulation of damage. It is an important data point to demonstrate the ability to collect long-run experimental data with our optimised data acquisition strategy, as it did not highlight limitations of these approach.

2) Add a description of curtaining over time, in response to the following point from reviewer 1: "curtains appear over time when milling, and it would be useful to understand how different sources behave over time in FIB/SEM tomography sessions. The score is currently done from individual windows milled, which gives a good indication of the performance. However, it would make sense to check that the behaviour remains identical in an imaging setting and with the moving milling windows (or lines). This will show the counteracting effect to the redeposition and etching effect reported when imaging with the E-beam the milled face. "

We have added a line in the manuscript to address the reviewer’s concern on page 4, lines 18 to 20 “During automated serial pFIB/SEM (presented on cells and tissues) we could not detect any trends in accumulation of curtaining or significant changes in curtaining propensity over time.” This is based on results presented in Author response image 1.

The primary source of curtaining is the presence of multiple surfaces with different milling rates. For life science samples working at cryogenic temperatures this could be the presence of ice contamination or specific compartments within the cell, such as a lipid droplet. Over the course of a serial pFIB/SEM run, each milling step increases the chance to encounter such an event. This chance could be more or less extreme depending on the combination of milling source and target material but is not linked to the length of time a sample is milled.

We have analysed (calculating the curtaining score for each slice of a given dataset) the rubrum, heart and Vero cells and did not find any specific trend towards higher rates of curtaining over time, thus demonstrating the stochastic nature of the phenomena.

**Author response image 1. sa2fig1:** Curtaining rate from the mouse heart, Vero cells and R. rubrum raw dataset as a function of slice number.

3) Add data on milling resolution and beam cross-section in response to the following point from reviewer 1: "No detail about the milling resolution has been reported. Since different currents and beams have different cross-sections, it is expected to affect the z-resolution achievable during an imaging session. It would be useful to have a description of the beam cross-sections at the various conditions used and how or whether these interfere with the preparation."

We have addressed this point by including an extra supplementary figure (Figure 1 – Supplementary Figure 5) which describes the profile of the beam for given currents and voltages for four plasma ion beam sources (N, O, Ar, Xe). We have also added a section on page 4 titles “plasma ion beam characterisation” which explains the results of these findings as well as the following note in the discussion relating to our extra work on page 12, lines 15-18:

“Importantly, the absolute size of the beam spot is not necessarily the limiting factor for milling resolution – fine control of the beam, careful placement of milling boxes and a steep beam profile can enable slice thickness or final lamella thickness smaller than the beam diameter. Therefore, for plasma ion beams it should be possible to ablate thicknesses below this value.”

The probe sizes were measured a few months after the experiments originally reported in this manuscript (so the apertures have been milled and therefore degraded). However, beam profiles are reported for measured current.

The theoretical probe sizes are shown in Author response image 2.

4) Comment on the effects of imaging tilt angle on collection efficiency.

A comment on these effects have been added on Page 6, Lines 33-35.

5) Explain the loss in resolution upon tilting the imaging angle, and the effect on contrast on the curtain edges (refer to reviewer 1 comment: "the generation of secondary electrons is known to increase with the increased tilt and to consider that the curtains (that are the prominent feature on the surface) are running along the tilt direction, it would be expected to see no contrast difference between the background and the edge of each curtain as the generation of secondary electrons will increase with tilt for both the edges and the background. Therefore, the contrast should be invariant, at least on the curtains").6) Comment on reviewer 1 claim of suboptimal astigmatism correction, and on how this might affect the comparison between the different imaging tilt angles.

Points 5 and 6 form variations on the same discussion, namely the effect of tilting the sample on image contrast. It is important to note that contrast and resolution are not the same thing; we have provided a sentence to highlight this. In our manuscript we discuss resolution of images.

We thank the reviewers for highlighting this distinction and agree with their comment. We would like to highlight that we have discussed this aspect (page 6, line 29 to line 34) and have presented data in Figure 4 – Supplementary Figure 1 that demonstrates this effect. To add extra weight to this aspect, we have moved this figure into Figure 4. We have edited the relevant paragraph (page 6, line 34 to 36) to describe how we determine the change in contrast levels between 52° and 90° images.

We have found it difficult to identify to which sub-optimal astigmatic image the reviewer was referring and could not identify astigmatism ourselves from the raw image. The presence of astigmatism in images is not dependent on the tilt angle but is a factor of tuning during SEM image acquisition. Tilting will not change the astigmatism correction. Every effort was used to try to keep the images as aberration free as possible during acquisition, but users may have found it hard to correct for these at first pass due to the low contrast used in images used to set up focus and astigmatism correction in initial set-up stages of serial runs. As a result, some images may have had sub-optimal astigmatism though this is not a trend (as hopefully visible throughout the many samples we analysed and resolution and segmentation analyses based upon them).

7) Discuss whether and how the contrast gained from the optimised approach using plasma FIB will allow interpretation and segmentation of biological features.Can automated segmentation approaches be applied to the FIB/SEM volumes in similar ways to cryo-ET reconstructions?

Automated segmentation strategies for cryogenic volumetric data are still in their infancy, in most cases requiring substantial manual training data and correction; and are often bespoke to the data under investigation. The area with most automation and the most successful outcomes is in segmentation of mitochondria. For this reason, we have focused on an automated segmentation of mitochondria within the serial pFIB/SEM heart tissue dataset. The methods used and evaluation of the results have been added to the manuscript as Figure 11, Results (Page 10, line 12-31, Discussion) (Page 13, line 13-19, and Materials and methods).

Could the authors demonstrate that the features detected in the SEM can be used to target areas of interest for cryo-ET? If so, can they show structural preservation in the cryo-ET reconstructions?

We thank the reviewer for their suggestion as this is indeed something we have pondered ourselves. To address this comment, we have performed correlative experiments incorporating fluorescence and electron tomography. As proposed by the referee we could show preservation of structures in cryoET and have updated the manuscript with the corresponding Results section “Combining pFIB/SEM with cryoCLEM and cryoET” (page 9 from line 1 to 22), Figure 9, the Materials and methods, Discussion (page 13, lines 1 to 12) and References.

Alternatively, the manuscript should more explicitly state that downstream correlative approaches are a potential application to be developed and tested in future work, and this is currently not possible.

We believe the above section should answer this reviewer comment.

Reviewer #2 (Recommendations for the authors):Suggestions for Improvement:The results presented in this manuscript are substantial on their own, and merit publication. However, as written, I worry that most readers will have a similar experience as me, reading excitedly, on the edge of our seats, waiting for the proof-of-concept that this approach is effective (and perhaps more effective than CLEM) for targeting and downstream cryo-ET imaging, and will come up somewhat disappointed that this is not demonstrated.I propose that the authors choose one of two possible paths to address the weakness above. The first, more extensive albeit thorough option, would be to address this by collecting tilt series data and reconstructing tomograms of the same lamella imaged by plasma cryo-FIB/SEM, and demonstrate that cross-correlation between volumetric cryo-FIB/SEM and cryo-ET acquisition is possible without affecting image/reconstruction quality. This could be evaluated either qualitatively (e.g. integrity of membranes/filaments) or structurally (e.g., by solving the subtomogram average of the cellular ribosome). Ideally, the authors would select a target and demonstrate an advantage of cryo-FIB/SEM targeting over the more standardized cryo-fluorescence/CLEM approaches.The second option would be to revise the manuscript text to more explicitly state upfront that this work is solely focused on evaluating the performance of cryo-FIB/SEM for volumetric imaging and ultrastructural analyses, and that future work will determine whether these are of sufficient quality to the image by cryo-ET. The authors sort of dance around the possibility that this approach could be used for downstream applications without sufficiently testing this, and therefore, I believe the intent of the article needs to be more explicitly addressed in the manuscript text. This would strengthen the merits of the article by allowing the reader to focus on the substantial improvements to contrast/imaging without feeling like something is "missing" at the end.Finally, I suggest that the authors apply an automated segmentation algorithm to their volumetric data as a more unbiased and practical metric to evaluate the degree to which cryo-FIB/SEM imaging increases resolution, contrast, and interpretability of subcellular features at scale that would enable biological insight.

We have collected preliminary data to demonstrate more concretely that cryogenic serial pFIB/SEM can be used as part of a correlative workflow with cryoET as a downstream step. However, as addressed above, optimisation of this workflow is outside of the scope of this manuscript. Careful analysis will be necessary to ensure that the high spatial frequencies require for high-resolution averaging techniques are not affected by decisions made during FIB/SEM data acquisition.